# Spatio-temporal control of mutualism in legumes helps spread symbiotic nitrogen fixation

Benoit Daubech[1†], Philippe Remigi[2†], Ginaini Doin de Moura[1], Marta Marchetti[1], Cécile Pouzet[3], Marie-Christine Auriac[1,3], Chaitanya S Gokhale[4], Catherine Masson-Boivin[1]*, Delphine Capela[1]

[1]The Laboratory of Plant-Microbe Interactions, Université de Toulouse, INRA, CNRS, Castanet-Tolosan, France; [2]New Zealand Institute for Advanced Study, Massey University, Auckland, New Zealand; [3]Fédération de Recherches Agrobiosciences, Interactions et Biodiversité, Plateforme d'Imagerie TRI, CNRS - UPS, Castanet-Tolosan, France; [4]Research Group for Theoretical Models of Eco-evolutionary Dynamics, Department of Evolutionary Theory, Max Planck Institute for Evolutionary Biology, Plön, Germany

**Abstract** Mutualism is of fundamental importance in ecosystems. Which factors help to keep the relationship mutually beneficial and evolutionarily successful is a central question. We addressed this issue for one of the most significant mutualistic interactions on Earth, which associates plants of the leguminosae family and hundreds of nitrogen ($N_2$)-fixing bacterial species. Here we analyze the spatio-temporal dynamics of fixers and non-fixers along the symbiotic process in the *Cupriavidus taiwanensis–Mimosa pudica* system. $N_2$-fixing symbionts progressively outcompete isogenic non-fixers within root nodules, where $N_2$-fixation occurs, even when they share the same nodule. Numerical simulations, supported by experimental validation, predict that rare fixers will invade a population dominated by non-fixing bacteria during serial nodulation cycles with a probability that is function of initial inoculum, plant population size and nodulation cycle length. Our findings provide insights into the selective forces and ecological factors that may have driven the spread of the $N_2$-fixation mutualistic trait.

DOI: https://doi.org/10.7554/eLife.28683.001

*For correspondence:
catherine.masson@inra.fr

†These authors contributed equally to this work

**Competing interests:** The authors declare that no competing interests exist.

## Introduction

The evolutionary dynamics of mutualistic interactions between higher organisms and microbes depends to a large extent on the transmission mode of microbial symbionts. Vertical transmission is expected to promote fitness alignment of obligate symbionts and their partners (*Herre et al., 1999*). In contrast, horizontal transmission generates more complex ecological cycles for facultative symbionts. When going through these cycles, microbes are subjected to several trade-offs regarding host range (specialist vs. generalist) and investment in the mutualism (good or bad cooperator, life in the host vs. outside the host). The large number of possible strategies to maximize fitness, and the ability to segregate in a population of genetically variable partners, often entails conflicts of interests between symbionts and their hosts (*Bever et al., 2009*; *Sachs et al., 2010*; *Porter and Simms, 2014*; *Jones et al., 2015*) that may result in the classic Tragedy of the Commons (*Hardin, 1968*). The emergence and stability of mutualism thus requires that proliferation of symbionts is allowed but restricted to appropriate spaces and times and that beneficial partners are ultimately favored over uncooperative ones (*Vigneron et al., 2014*; *Visick and McFall-Ngai, 2000*; *Koch et al., 2014*). The theoretical aspects of the evolution and maintenance of mutualistic

**eLife digest** Rhizobia are soil bacteria that are able to form a symbiotic relationship with legumes – plants that include peas, beans and lentils. The bacteria move into cells in the roots of the plant and cause new organs called nodules to form. Inside the nodules the bacteria multiply before being released to the soil again. Also while in the nodules, the bacteria receive carbon-containing compounds from the plant. In return many of the bacteria convert (or "fix") nitrogen from the air into compounds that the plant can use to build molecules such as DNA and proteins. Yet, some of the bacteria are "non-fixers" that provide little or no benefit to the host plant.

Evidence suggests that legumes select against non-fixer bacteria, though it was not clear when or how this selection process occurs. Daubech, Remigi et al. have now followed the number and viability of two variants of a bacteria species called *Cupriavidus taiwanensis* as they form a symbiotic interaction with *Mimosa pudica*, a member of the pea family. The two types of bacteria differed only by whether or not they were able to fix nitrogen. At first fixers and non-fixers entered nodules and multiplied at equal rates. Later, the fixers progressively outcompeted the non-fixers. Then, around 20 days after the bacteria entered the plant, nodule cells that contained non-fixers degenerated. This indicates that the nodule cells help to control bacterial proliferation based on the benefits they receive in return.

Further experiments and mathematical modeling also showed that over repeated cycles of root nodule formation, nitrogen fixers can invade a bacterial population dominated by non-fixer bacteria. The likelihood that this invasion will be successful increases as three other factors increase: the proportion of fixer bacteria in the initial population, the number of available plants, and the length of time the bacteria spend in the nodules. This mechanism ensures the maintenance and spread of nitrogen-fixing traits in the bacterial population.

Improving the processes of biological nitrogen fixation could help to reduce the amount of fertilizers required to grow crops. This in the future could help make agricultural ecosystems more sustainable. The results presented by Daubech, Remigi et al. provide guidelines that could be used to select nitrogen-fixing bacteria on legume crops or on nitrogen-fixing cereals that may be engineered in the future. Further work is now needed to understand in more detail the molecular mechanisms that lead to the death of non-fixer bacteria.

DOI: https://doi.org/10.7554/eLife.28683.002

interactions have been extensively discussed (*Archetti et al., 2011*; *Akcay, 2015*). Yet experimental assessment is scarce and the impact of ecological factors, such as population size of hosts and symbionts or the duration of the interaction, has been under-explored, although they are an essential component of the evolutionary potential of symbiotic systems.

Rhizobia, the $N_2$-fixing symbionts of legumes, induce the formation of and massively colonize nodules, where intracellular bacteria fix atmospheric nitrogen for the benefit of the plant in exchange for photosynthates. When the nodule senesces, nodule bacteria are released to the soil where they can return to free-living lifestyle and/or colonize a new host (*Thies et al., 1995*). During evolution, symbiosis modules carrying genes essential for the symbiotic process have spread to many different taxa so that extant rhizobia are distributed in hundreds of species in 14 genera of α- and β-proteobacteria (*Remigi et al., 2016*). Acquisition of symbiotic genes may not be sufficient to create an effective symbiont and may lead to bacteria exhibiting various levels of symbiotic capacities (*Nandasena et al., 2006*; *Nandasena et al., 2007*; *Marchetti et al., 2010*) that can be further optimized and maintained under legume selection pressure (*Marchetti et al., 2017*; *Marchetti et al., 2014*). It has been established that bacteria better able to form and infect nodules are selected by a partner choice mechanism involving the specific recognition of bacterial molecular signals by plant receptors (*Kawaharada et al., 2015*; *Radutoiu et al., 2003*). Bacterial features that are recognized by the plant include Nod factors that initiate rhizobial entry and nodule formation (*Perret et al., 2000*; *Broghammer et al., 2012*), and lipo/exopolysaccharides critical for root infection and bacterial release inside the plant cell (*Kawaharada et al., 2015*), as well as an array of bacterial effectors that refine host specificity (*Deakin and Broughton, 2009*). Nitrogen fixation however is uncoupled from nodulation and infection, and legumes can be nodulated and infected by

ineffective symbiotic partners (*Gehlot et al., 2013*; *Gourion et al., 2015*). The emergence of mutualism in populations resulting from the transfer of symbiosis modules, and its maintenance over evolutionary timescales (*Werner et al., 2014*) indicates that the cooperative behaviour of the bacterial symbionts is controlled at the infection and/or post-infection levels by one or a combination of mechanisms. Partner choice is the selection of appropriate symbionts at the (pre-) infection stage based on signal recognition while post-infection sanctions rely on the ability to discriminate between low- and high-quality cooperators during an established interaction and to punish or reward them accordingly (*Kiers and Denison, 2008*; *Frederickson, 2013*). Partner-fidelity feedback (PFF) ensures positive assortment of symbionts during long lasting or repeated interactions in spatially structured environments independently from any recognition process or conditional response (*Sachs et al., 2004*). These different control mechanisms have been proposed to affect the dynamics of mutualistic traits, particularly in the context of the nitrogen-fixing symbiosis (*Kiers et al., 2003*; *Oono et al., 2009*). Here we evaluate how selective forces and ecological factors act on the dissemination of the nitrogen fixation mutualistic trait on the *Cupriavidus taiwanensis-Mimosa pudica* mutualistic interaction. Specifically we evaluated the spatio-temporal dynamics of $N_2$-fixing and non-fixing bacterial subpopulations to model the spread of the $N_2$-fixation trait across plant generations.

## Results

### Evidence for a spatial and temporal control of mutualism in *Mimosa* nodules

During the symbiotic process, most rhizobia enter the legume root via infection threads that ensure colonization of the forming nodule and ultimately release bacteria into nodule cells where differentiated forms called bacteroids fix nitrogen (*Batut et al., 2004*). Although they induce the formation of indeterminate nodules, it is noteworthy that *Cupriavidus taiwanensis* symbionts of *Mimosa* spp. are not terminally differentiated and ca. 20% of bacteroids recovered from nodules, together with bacteria present in infection threads, can resume growth (*Marchetti et al., 2011*). To evaluate the specific fates of mutualists and non mutualists in plants infected by a mixed population, we monitored the fitness of total nodule bacteria over time following co-inoculation of *Mimosa pudica* seedlings with a mixture (1/1 ratio, $10^6$ total bacteria/plant) of isogenic $N_2$-fixing and non-fixing strains of *C. taiwanensis*. Fix$^+$ and Fix$^-$ strains only differed by the presence of the *nifH* gene, encoding the nitrogenase reductase subunit of the nitrogenase enzyme, and of constitutively expressed GFP or mCherry fluorescent genes. For technical reasons (see Materials and methods), nodules were only collected from 14 dpi. Importantly, each nodule was individually analyzed for bacterial fitness by plating, allowing analysis at the nodule and plant individual levels. In these experimental conditions 97% of the nodules were infected by either Fix$^+$ or Fix$^-$ bacteria.

We observed a marked difference in the reproductive fitness of Fix$^+$ and Fix$^-$ bacteria from the same plant over time, which significantly differed from 21 days post-infection (dpi) and up to 28 fold on average (*Figure 1A* and *Figure 1—figure supplement 1*), perhaps because of plant control mechanisms, including sanctions (*Kiers et al., 2003*) and possibly PFF. A significant difference was also obtained from 28 dpi when analyzing control plants singly-infected with either Fix$^+$ or Fix$^-$ strains (*Figure 2A*). Non-fixers did not proliferate better than fixers even at 14 dpi (*Figure 1A*) possibly because the metabolic cost paid by bacteria to fix nitrogen in terms of ATP and reducing power is too low to be detected in our experimental conditions, or because plant sanctions/PFF and the metabolic cost of nitrogen fixation equilibrate until sanctions become prominent. The resulting net fitness cost of cooperation, which is the weighted metabolic cost of nitrogen fixation by any form of plant control, thus appeared to be zero or negative, enabling mutualism to spread.

The differential fitness was not due to a better nodulation competitveness of Fix$^+$ bacteria. The number of nodules formed by each strain was indeed proportional to the inoculum ratio (1/1) throughout the time course (*Figure 3*), confirming that bacterial nitrogen-fixing ability is not selected at the root entry level (*Hahn and Studer, 1986*; *Westhoek et al., 2017*). Yet the number of nodules in nitrogen-starved non-fixing plants (infected with 99% or 100% Fix$^-$) constantly increased over a 42 day period, while this number reached a plateau at *ca.* 20 dpi in healthy $N_2$-fixing plants (infected with 50% or 100% Fix$^+$) (*Figure 4*), indicative of a mechanism of autoregulation of nodulation acting at the whole-plant level (*Ferguson et al., 2010*) and depending on the nitrogen status of the plant

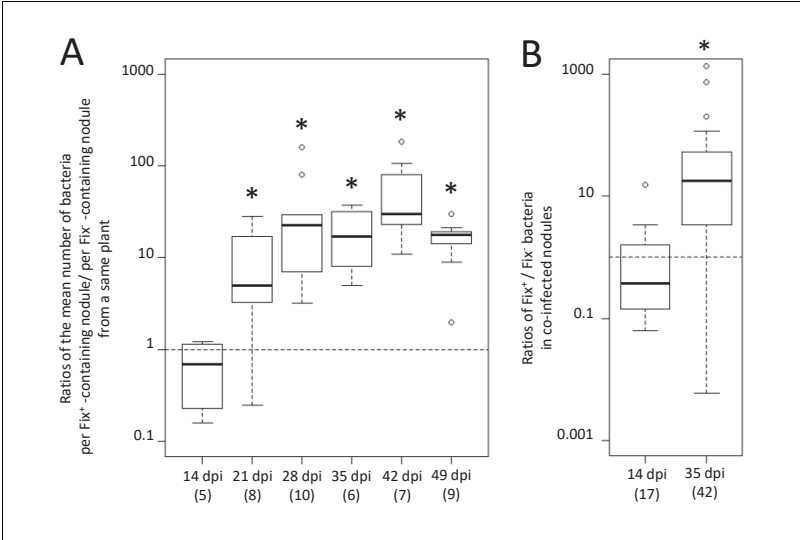

**Figure 1.** Kinetics of reproductive fitness of Fix$^+$ and Fix$^-$ bacteria in nodules following co-inoculation of *M. pudica*. *M. pudica* plants were co-inoculated with a mixture of Fix$^+$ and Fix$^-$ strains at a 1/1 ratio, using $10^6$ (**A**) or $10^{10}$ bacteria/plant (**B**). Nodules were individually analyzed by plating their bacterial population (see *Figure 1—figure supplement 1*). Co-infected nodules represented ca. 3% (**A**) or 20% (**B**) of the nodules. (**A**) The ratio of the mean number of bacteria per Fix$^+$-containing nodule to the mean number of bacteria per Fix$^-$- containing nodule was calculated for each individual plant at each time point (see *Figure 1—figure supplement 1*) and box plots represent the distribution of these ratios (*Figure 1—source data 1*). Only single-infected nodules were taken into account in this graph. (**B**) Box plots represent the distribution of the ratios of Fix$^+$ bacteria to Fix$^-$ bacteria in co-infected nodules (*Figure 1—source data 2*). Central rectangles span the first quartile to the third quartile (that is, the interquartile range or IQR), bold segments inside rectangles show the median, unfilled circles indicate suspected outliers, whiskers above and below the box show either the locations of the minimum and maximum in the absence of suspected outlying data or 1.5 × IQR if an outlier is present. Horizontal dashed lines correspond to ratios equal to 1. The number of plants (**A**) or nodules (**B**) analyzed is indicated in brackets. *Significant differences between the number of Fix$^+$ and Fix$^-$ bacteria per nodule ($p<0.05$, multiple comparison test after Kruskal-Wallis (**A**); $p<0.001$, after Student t-test with paired data (**B**).

DOI: https://doi.org/10.7554/eLife.28683.003

The following source data and figure supplement are available for figure 1:

**Source data 1.** Reproductive fitness of nodule bacteria following co-inoculation with Fix$^+$ (CBM2700) and Fix$^-$ (CBM2707) *C. taiwanensis*.
DOI: https://doi.org/10.7554/eLife.28683.005

**Source data 2.** Reproductive fitness of nodule bacteria in nodules co-infected by Fix$^+$ (CBM2700) and Fix$^-$ (CBM2707) *C. taiwanensis*.
DOI: https://doi.org/10.7554/eLife.28683.006

**Figure supplement 1.** Kinetics of reproductive fitness of nodule bacteria following co-inoculation with Fix$^+$ (CBM2700) and Fix$^-$ (CBM2707) *C. taiwanensis*.
DOI: https://doi.org/10.7554/eLife.28683.004

---

(*Malik et al., 1987*; *van Noorden et al., 2016*). This difference in time course increases the chance that a rare Fix$^+$ among a Fix$^-$ population will form a nodule .

To identify the spatial level at which selection applies we first analyzed double occupancy nodules, which were obtained in significant proportion by modifying the plant culture system and increasing the inoculum density by four logs (see Materials and methods). Co-infected nodules contained a similar number of Fix$^+$ and Fix$^-$ bacteria at 14 dpi, but on average ca. 80 times more N$_2$-fixing bacteria than non-fixing bacteria at 35 dpi (*Figure 1B*), indicating that the control occurs at the nodule scale. Previous studies established that bacteroids do not persist in nodule cells of nitrogen-starved plants infected only by non-fixers, leading to premature nodule senescence (*Berrabah et al., 2015*; *Hirsch and Smith, 1987*), while they persist in healthy plants singly-infected with fixers. We therefore then analyzed the viability of bacteroids on sections of singly-occupied or double-occupied nodules collected from co-inoculation experiments using propidium iodide (PI), which stains dead

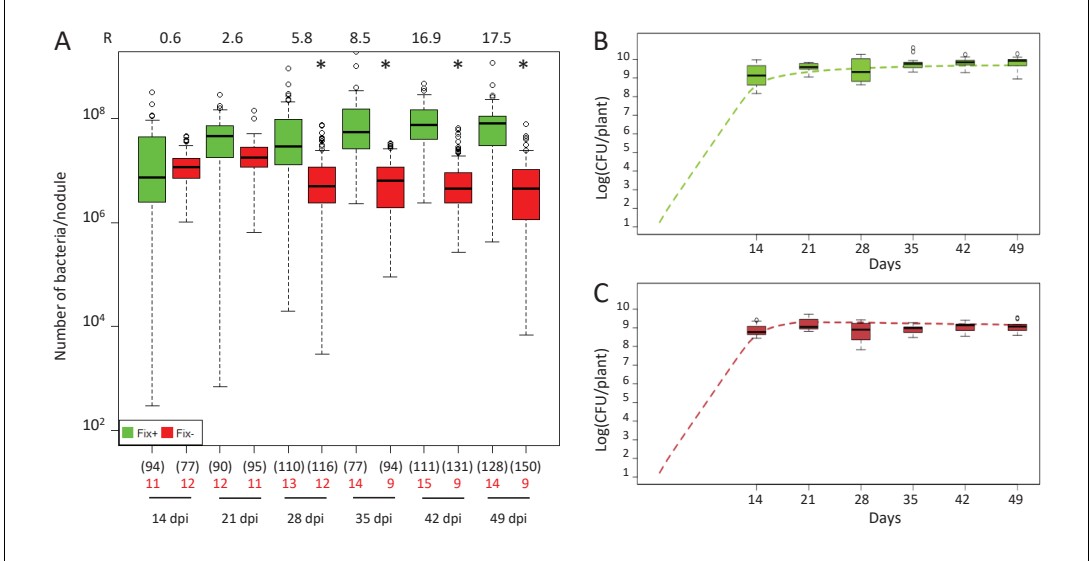

**Figure 2.** Kinetics of reproductive fitness of Fix+ or Fix- nodule bacteria following single-inoculation of *M. pudica*. (A) Fix+ (CBM382) or Fix- (CBM2568) *C. taiwanensis* were inoculated on *M. pudica*. Box plots represent the distribution of the number of bacteria recovered per nodule on plates. Box plots were constructed as described in *Figure 1*. R, ratios of the median number of Fix+ bacteria per nodule on the median number of Fix- bacteria per nodule. The number of nodules analyzed at each time point is indicated in brackets. The number of plants analyzed at each time point is indicated in red. Results are from two independent experiments (*Figure 2—source data 1*). *Significantly different from the number of Fix+ bacteria per nodule (p<0.05 multiple comparison test after Kruskal-Wallis). (B, C) Theoretical reproductive fitness of Fix+ (B) and Fix- bacteria (C) following single-inoculation of *M. pudica* as compared to experimental data. Dotted lines represent bacterial populations per plant averaged over 200 replicate simulations (*Figure 2—source data 2*). Box plots represent the distribution of the number of bacteria experimentally recovered per plant. Experimental data are from (A).

DOI: https://doi.org/10.7554/eLife.28683.007

The following source data is available for figure 2:

**Source data 1.** Reproductive fitness of nodule bacteria following single-inoculations with either Fix+ (CBM382) or Fix- (CBM2568) *C. taiwanensis*.
DOI: https://doi.org/10.7554/eLife.28683.008

**Source data 2.** Simulation data for the reproductive fitness of Fix+ and Fix- bacteria following single inoculations of *M. pudica*.
DOI: https://doi.org/10.7554/eLife.28683.009

cells (*Virta et al., 1998*). Bacteroid viability in Fix+-occupied nodules remained stable from 14 to 35 dpi (*Figure 5D*). By contrast, bacteroids in the nitrogen-fixing zone of Fix--occupied nodules started losing viability at 16–21 dpi and were all dead (PI-stained) at 35 dpi (*Figure 5E*). Electron microscopy confirmed signs of nodule cell and bacterial degeneration in Fix--occupied nodules at 19 dpi (*Figure 6*). Co-infected nodules showed clear sectoring, with infected plant cells in one part filled with Fix+ strains and in the other part filled with Fix- strains (*Figure 5FGHI*). We never observed co-infected nodule cells. While at 14 dpi both strains were alive (*Figure 5G*), at 35 dpi only Fix- bacteroids were PI-stained confirming that Fix+ and Fix- intracellular bacteria have distinct fates within the same nodule (*Figure 5HI*). The ca. $5 \times 10^6$ bacteria recovered at 35 dpi from nodules infected with only Fix- bacteria may thus be bacteria colonizing the infection threads and the infection zone.

In conclusion we provide evidence for differential spatio-temporal dynamics of $N_2$-fixing and non-fixing partners during the symbiotic process, highlighting the importance of considering temporal variations when studying the evolution of cooperative interactions (*Barker and Bronstein, 2016*). We established that the control of mutualism (i) acts at the nodule cell scale, (ii) occurs relatively early, ca. 16–21 days after inoculation when the wild-type nitrogenase is fully active in Fix+ bacteria (*Figure 7*) and (iii) leads to up a ca. 80 fold relative increase in mutualistic partners.

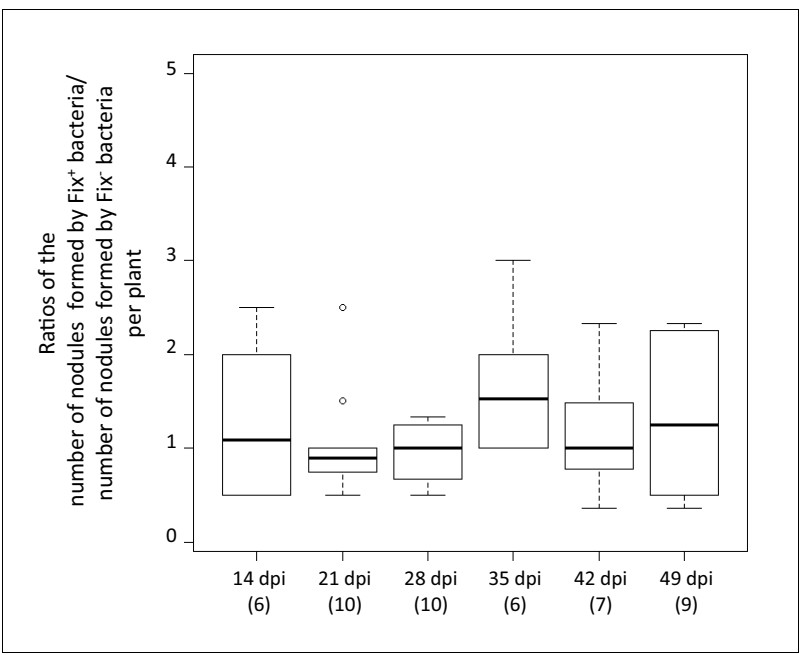

**Figure 3.** Relative number of nodules formed by Fix$^+$ and Fix$^-$ bacteria per plant individual. *M. pudica* plants were co-inoculated with the CBM2700 (Fix$^+$, GFP) and CBM2707 (Fix$^-$, mCherry) strains at a 1/1 ratio. The number of plants analyzed for each time point is indicated in brackets. Boxplots were constructed as described in *Figure 1*. No significant differences were observed between the number of nodules formed by Fix$^+$ bacteria and Fix$^-$ bacteria per plant at the different time points ($p > 0.05$, Student *t*-test with paired data at each time point or multiple comparison test after Kruskal-Wallis on the whole dataset) (*Figure 3—source data 1*).
DOI: https://doi.org/10.7554/eLife.28683.010

The following source data is available for figure 3:

**Source data 1.** Relative number of nodules formed by Fix$^+$ and Fix$^-$ bacteria per plant individual.
DOI: https://doi.org/10.7554/eLife.28683.011

## Eco-evolutionary dynamics of N$_2$-fixers and non-fixers through serial nodulation cycles

Next, we addressed the question of whether mutualism control will allow a minority Fix$^+$ subpopulation to invade the symbiotic population.

We first used our experimental data to develop a stochastic mathematical model qualitatively simulating the fate of *C. taiwanensis* populations during nodulation in *M. pudica* plants. The two key components of this model are (i) the kinetics of nodule formation from bacteria randomly chosen from the rhizospheric population and (ii) bacterial multiplication within nodules, according to bacterial genotype (see Materials and methods and *Table 1* for details on model construction and parameterization). While the model is developed as a proof-of-concept, instead of a simple deterministic model we chose to include stochasticity in the nodulation process in order to reflect the variability observed in the experimental data. In order to test our model, we first simulated the reproductive fitness of nodule bacteria following single-inoculation with either Fix$^-$ or Fix$^+$ bacteria over a 49 day-period, and compared this simulation to the kinetics experimentally observed (*Figure 2BC*). We then both simulated and experimentally determined the relative proportion of Fix$^+$ bacteria recovered from plants co-inoculated with a minor subpopulation of Fix$^+$ (1%) and a major subpopulation of Fix$^-$ (99%) bacteria over 49 days (*Figure 8*). Simulation outcomes qualitatively matched the dynamics of bacterial populations observed experimentally (*Figures 2BC* and *8*), indicating that the experimentally measured and inferred model parameters are appropriate for studying the evolutionary dynamics of *C. taiwanensis* populations in different ecological conditions.

We then used this model to explore how plant population size and the length of inoculation cycles impact on the dynamics of *C. taiwanensis* populations during serial cycles of inoculation of *M. pudica* plants and re-isolation of bacteria from nodules. Starting with a fixed proportion of Fix$^+$

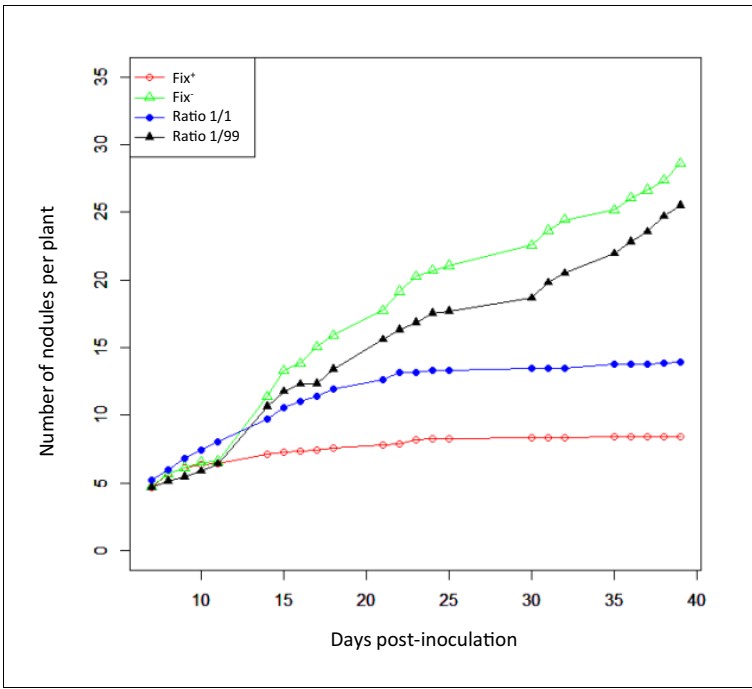

**Figure 4.** Nodulation kinetics. *M. pudica* plants were single-inoculated with either CBM832 (Fix⁺) or CBM2568 (Fix⁻) or co-inoculated with a mixture of both strains at a 1/1 or 1/99 ratio. First nodules appeared at 5–7 dpi (*Figure 4—source data 1*).

DOI: https://doi.org/10.7554/eLife.28683.012

The following source data is available for figure 4:

**Source data 1.** Nodulation kinetics of Fix⁺ (CBM382) and Fix⁻ (CBM2568) *C. taiwanensis* following single- or co-inoculation of *M. pudica*.

DOI: https://doi.org/10.7554/eLife.28683.013

bacteria (1% or 0.1%) in the inoculum, we varied the number of inoculated plants from 1 to 100 (or 1 to 1000) and the length of nodulation cycles (time from plant inoculation to nodule bacteria harvesting) from 14 to 49 days, which is shorter than the lifespan of a nodule in nature. We found that larger plant pools and longer cycles progressively reduced extinction probabilities and increased the proportion of Fix⁺ in the nodule bacterial population (*Figure 9A* and *Figure 9—figure supplement 1*). For example, the model predicted that using an initial inoculum of 1% Fix⁺, 4 cycles of 42 days with pools of 20 plants were sufficient to yield more than 85% of Fix⁺ bacteria in all replicates where Fix⁺ populations avoided extinction (89 times out of 100 replicates in *Figure 9A*). Smaller plant pools or shorter cycles all yielded higher probabilities of extinction and decreased proportions of Fix⁺ bacteria. An initially lower Fix⁺ proportion (0.1%) could be compensated for by a higher plant population size and/or a longer cycle length (*Figure 9—figure supplement 1*). We analyzed in detail the dynamics of Fix⁺ subpopulations over 10 cycles in a situation where the cycle length had a major impact on the evolutionary outcome (20 plants) (*Figure 9A*) and plotted the proportion of Fix⁺ bacteria recovered after each cycle, for cycles ranging from 14 to 49 days (*Figure 9B*). We observed that, in the vast majority of cases, the fate of Fix⁺ populations is already determined after the first cycle: these populations are either bound to extinction (with a probability indicated in *Figure 9A*) or to a gradual increase in frequency that ultimately leads to fixation. This result holds true for all cycle lengths except 14 days, where population dynamics is dominated by drift due to the equivalent fitness of Fix⁻ and Fix⁺ clones (*Figure 1A*). A key factor controlling the early bifurcation between extinction and fixation of Fix⁺ population is the probability that a Fix⁺ bacterium forms a nodule during the first cycle, which depends on both the size of plant pools and the length of nodulation cycles.

Understanding the influence of plant pool size is straightforward. Very few nodules are produced on each plant, creating a bottleneck in bacterial population size at each nodulation cycle. Whatever

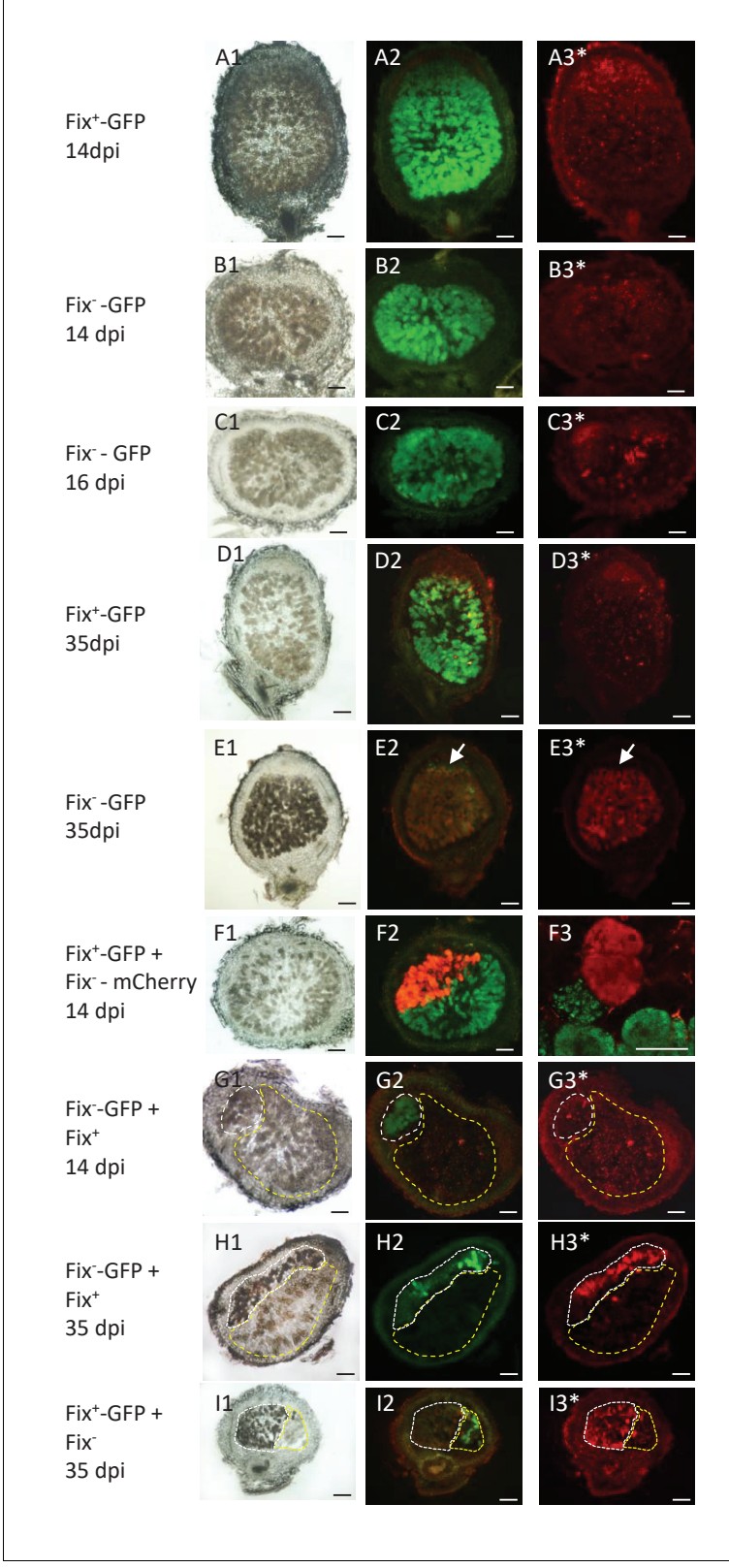

**Figure 5.** Viability of Fix+ and Fix- bacteroids. *M. pudica* were co-inoculated with Fix+ and Fix- *C. taiwanensis* at a 1/1 ratio and sections of nodules collected at 14 dpi (ABFG), 16 dpi (**C**) or 35 dpi (DEHI) were observed under bright field (panels 1) or fluorescent microscopy (panels 2 and 3), and after PI staining (panels with an *). Panels with the same letters represent the same nodule section. (**F3**), magnification of (**F2**) visualized by confocal

*Figure 5 continued on next page*

*Figure 5 continued*
microscopy. (**A**) and (**D**), sections of nodules infected with a GFP-labeled Fix+ strain. (**B**) (**C**) and (**E**), sections of nodule infected with a GFP-labeled Fix- strain. (**F**), nodule co-infected with a GFP-labeled Fix+ and a mCherry-labeled Fix- strain. (**G**) and (**H**), nodules co-infected with a GFP-labeled Fix- and an unlabeled Fix+ strain. (**I**), nodules co-infected with a GFP-labeled Fix+ and an unlabeled Fix- strain. The white and yellow dotted lines in (GHI) delimit the areas occupied by the Fix- and Fix+ strains in a co-infected nodule, respectively. Note that neither the Fix+ (**D3**) nor the Fix- bacteroids (**B3G3**) are red-labeled by PI staining at 14 dpi whereas a few cells are PI-stained in the Fix--occupied nodule at 16 dpi ([**C3**], arrows), and Fix- are mostly PI-labeled (dead) at 35 dpi (**E3H3I3**). Note that bacteria of the infection zone are still alive at 35 dpi (arrow, **E2E3**). Note that nodule cells filled with Fix- are browner than nodule cells filled with Fix+ (**G1H1I1**). Scale bars correspond to 100 µm except for **F3** (30 µm).
DOI: https://doi.org/10.7554/eLife.28683.014

the cycle length, larger numbers of plants per pool increase the likelihood that at least one Fix+ clone is sampled from the rhizospheric population, giving Fix+ subpopulations an opportunity to increase in frequency and avoid extinction in the next cycle (*Figure 9A* and *Figure 9—figure supplement 1*). Under longer cycles, extinction probability decreases (*Figure 9AB*) since more nodules are produced (*Figure 4*) and the size of Fix+ populations increases at a faster rate (*Figure 9B*) as a result

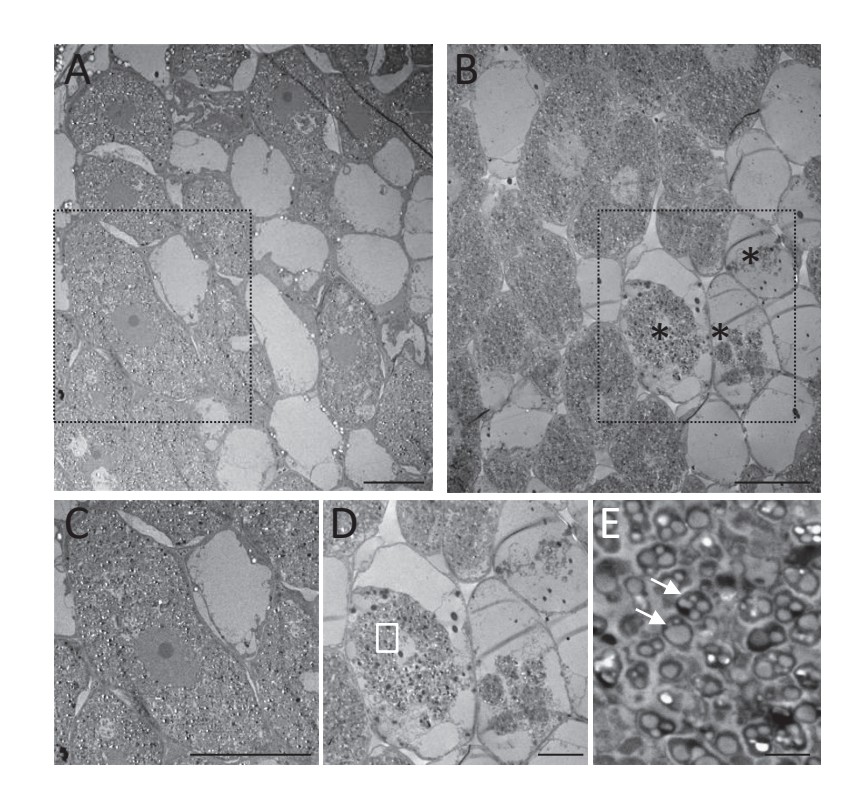

**Figure 6.** Electron microscopy of Fix+- and Fix--occupied nodules. *M. pudica* plants were co-inoculated with Fix+ (CBM2708, mCherry) and Fix- (CBM2568, unlabeled) *C. taiwanensis* at a 1/1 ratio. Nodules collected at 19 dpi (ABCDE) were sorted for mCherry expression under fluorescence microscopy and used for electron microscopy observation. Degenerated nodule cells (*) were observed in Fix--occupied nodules (BDE) but not in Fix+-occupied nodules (AC). (**C**) and (**D**) represent magnification of the zones delimited by a black dashed rectangle in (**A**) and (**B**) respectively. (**E**) magnification of the white rectangle in (**D**) showing degenerated bacteria (arrows). Scale bars represent 20 µm (ABC), 10 µm (**D**) and 2 µm (**E**).
DOI: https://doi.org/10.7554/eLife.28683.015

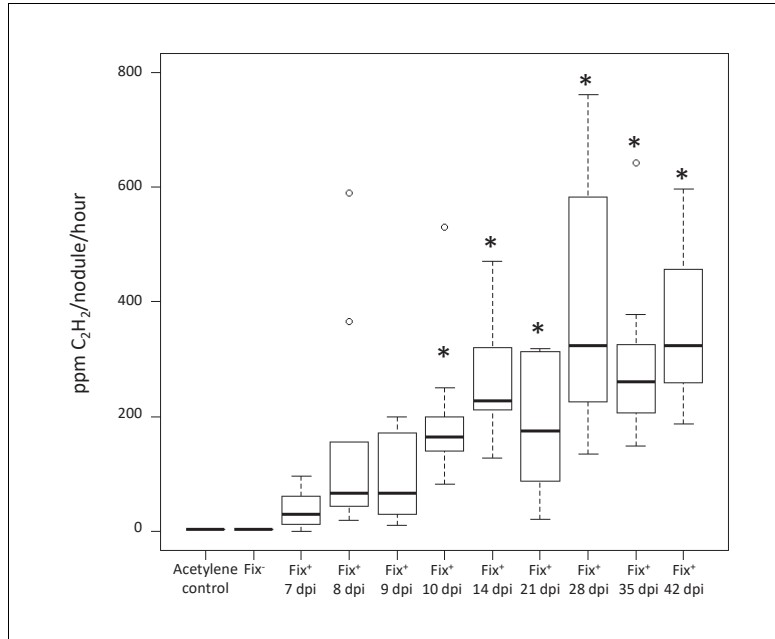

**Figure 7.** Kinetics of nitrogenase activity in $N_2$-fixing *M. pudica* nodules. Plants were inoculated with *C. taiwanensis* CBM832 (Fix[+]), and nitrogenase activity measured using the acetylene reduction assay (ARA) (*Figure 7—source data 1*). Two negative controls, *i.e.* tubes containing only the acetylene substrate and plants inoculated with *C. taiwanensis* CBM2568 (Fix[-]), were included. In these cases, boxplots correspond to data from all time points. *, Significantly different from the negative controls (p<0.05 after multiple comparison test of Kruskal-Wallis).
DOI: https://doi.org/10.7554/eLife.28683.016
The following source data is available for figure 7:

**Source data 1.** Nitrogenase activity of *C. taiwanensis* Fix[+] (CBM832).
DOI: https://doi.org/10.7554/eLife.28683.017

---

of a decrease in Fix[-] fitness in older nodules (*Figure 1A*). The combined action of these two factors act on the inoculum for next cycle, generating an eco-evolutionary feedback.

To assess the predictions of the model experimentally, we performed serial inoculation-nodulation cycles of 21 or 35 days using 20 *M. pudica* plants and an initial inoculum of $5 \times 10^3$ Fix[+]/$5 \times 10^5$ Fix[-] *C. taiwanensis* per plant. In each 35 day-cycle the nitrogen-fixing subpopulation increased and it reached nearly 100% of the population after four cycles (*Figure 10A*), similar to what observed with the model. Under 21 day-cycles, both simulations and experiments lead to a slower progression of Fix[+] subpopulations (*Figure 10B*). It is worth noting that an increase in frequency of the best cooperators among natural strains was also observed after three consecutive nodulation cycles between *Medicago truncatula* and *Sinorhizobium meliloti* (*Heath and Tiffin, 2009*), indicating that the selective advantage of the best $N_2$-fixing strains seems to be robust to the natural diversity of symbiotic associations.

## Discussion

Identifying the selective forces and ecological factors that shape mutualism is central to predicting its maintenance and dissemination over evolutionary scales. Here we provide conclusive evidence that nitrogen fixation per se, the ultimate trait that turns a parasitic rhizobium-legume association into a mutualistic one, determines the *in planta* spatio-temporal fate of endosymbiotic bacteria. Non-$N_2$-fixing symbionts do not persist within cells of indeterminate nodules even when they share a nodule with $N_2$-fixing symbionts, indicative of a cell autonomous senescence program as recently shown for determinate nodules (*Regus et al., 2017*). This results in the progressive and selective *in planta* expansion of fixers during the symbiotic process.

**Table 1.** Model parameters

| Parameter | Abbreviation | Value |
|---|---|---|
| Size of each pool of plants[*] | Pool | Variable (1–1000) |
| Number of replicates[*] | Rep | Variable (5 or 100) |
| Length of each cycle[*] | Days | Variable (14-49) |
| Number of cycles[*] | Cyc | Variable (4 or 10) |
| Initial proportion of Fix[+] cells[*] | x | Variable (1 or 0.1) |
| Maximum number of new nodules/plant/day[†] | $\lambda_{max}$ | 0.44 |
| Coefficient for the auto-regulation of nodulation in nodulation kinetics[†] | $a_1$ | 0.03 |
| Coefficient for time-decay in nodulation kinetics[†] | $a_2$ | 0.006 |
| Lag for time-decay in nodulation kinetics[†] | $a_3$ | 2 |
| Growth rate of bacteria within nodule[†] | r | 1.95 |
| Fitness cost of nitrogen fixation[‡] | c | 0 |
| Sanctions for Fix[-][‡] | s | 1.65 |
| Day at which additional sanctions begin[‡] | ds | 17 |
| Nodule carrying capacity[‡] | K | $1.4 \times 10^8$ |

[*] parameters varied in the simulations
[†] experimentally measured parameters
[‡] parameters inferred from experimental data
DOI: https://doi.org/10.7554/eLife.28683.021

The most likely explanation is that the plant exerts a post-infection control of $N_2$-fixation that overcomes the metabolic cost of nitrogen fixation paid by mutualistic bacteria. Sanctions could occur as defense responses and/or by decreasing nutrient supply to non-fixing bacteroids. Given that Fix[-] and Fix[+] bacteria are spatially segregated within nodules, the latter case could also result from the local degeneration of nodule cells, and be interpreted as an example of Partner Fidelity-Feedback mechanism occurring at the level of individual cells (*Shou, 2015*). Since control mechanisms prevent social dilemma –*i.e.* the possibility that one partner increases its own fitness by decreasing its investment in mutualism- and help cooperation persist (*Kiers and Denison, 2008*; *Frederickson, 2013*; *Sachs et al., 2004*), non-fixers do not threaten mutualism in our system. Yet the fate of strains able to fix intermediate levels of nitrogen fixation may be different. Monitoring the fitness of strains varying in their nitrogen fixation capacity would provide a more complete picture of mutualism control. Nevertheless, our results provide an additional example supporting the emerging idea that low quality rhizobial partners rarely benefit from low investment in mutualism (*Jones et al., 2015*; *Friesen, 2012*). Plant sanctions resulting in bacterial fitness reduction were demonstrated in some rhizobium-legume systems by simulating $N_2$ deficiency via gas manipulation around nodules (*Kiers et al., 2003*; *Oono et al., 2011*), although not seen in other systems (*Marco et al., 2009*; *Ling et al., 2013*). That different plants may rely on different control mechanisms would not be surprising given the variety of mechanisms that lead to symbiosis with legumes (*Masson-Boivin et al., 2009*).

Experimental investigations can fuel a theoretical framework able to reframe general evolutionary questions in an ecological context (*Hoek et al., 2016*). Our qualitative model of the eco-evolutionary dynamics of mutualistic and non-mutualistic populations includes serial inoculation-nodulation cycles. This regime mimics an experimental set up of horizontal transmission of rhizobia across plant generations albeit on an accelerated basis. A general outcome of the model is that rare fixers will invade a population dominated by non-fixing bacteria, above a threshold combination of plant and bacterial population sizes and cycle lengths. The model helps explore further combinations of number of cycles, cycle lengths and plant pool sizes to hypothesize the evolutionary trajectory of the Fix[+] genotype. While the selective advantage of the Fix[+] phenotype is expected to ensure its fixation in a deterministic manner, strong population bottlenecks occurring at the nodulation step introduce a source of stochasticity in these dynamics and may thus prevent the action of directional selection. The effect of stochasticity has been shown to be of immense evolutionary consequence in related

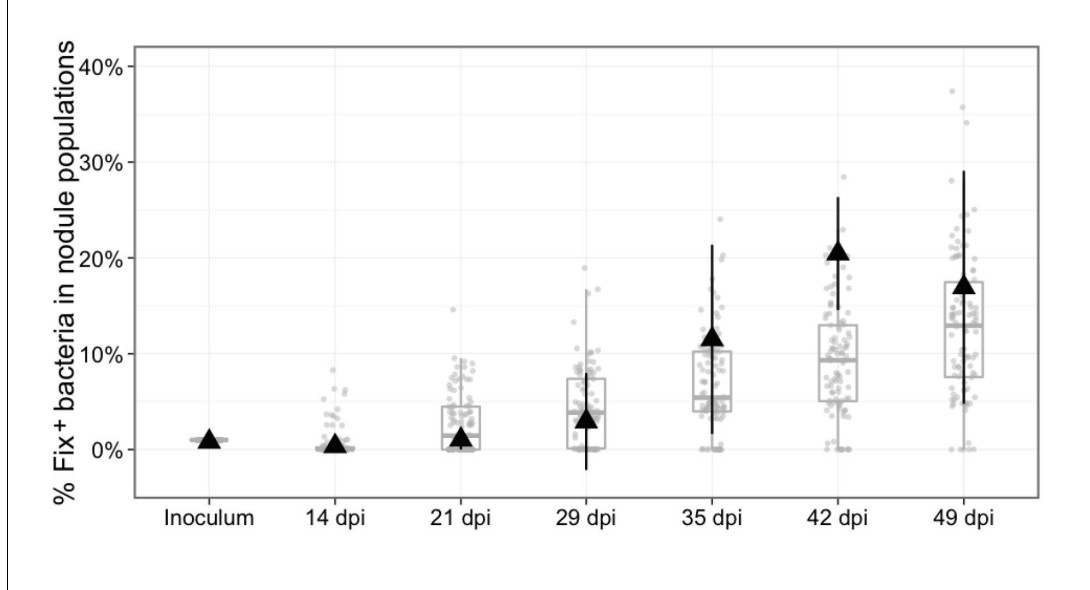

**Figure 8.** Experimental and theoretical reproductive fitness of Fix[+] and Fix[-] bacteria following co- inoculation of *M. pudica* (ratio 1/100). The proportion of Fix[+] clones in nodules was experimentally measured and simulated over 49 days, following co-inoculation of 20 plants. Experimental data are shown as black triangles (*Figure 8—source data 1*). Black error bars represent standard deviation from 2 to 3 replicates. The results from 100 replicate simulations are shown as grey dots and boxplots (*Figure 8—source data 2*).

DOI: https://doi.org/10.7554/eLife.28683.018

The following source data is available for figure 8:

**Source data 1.** Experimental data for the reproductive fitness of Fix[+] and Fix[-] bacteria following co- inoculation of *M. pudica* (ratio 1/100) over 49 days.
DOI: https://doi.org/10.7554/eLife.28683.019
**Source data 2.** Simulation data for the reproductive fitness of Fix[+] and Fix[-] bacteria following co- inoculation of *M. pudica* (ratio 1/100) over 49 days.
DOI: https://doi.org/10.7554/eLife.28683.020

models of host parasite coevolution (*Papkou et al., 2016*). Another characteristic of our system is that, when the Fix[+] populations increase in abundance then so does their proliferation, leading to a quick increase of Fix[+] over successive nodulation cycles (*Figure 9B*). This interaction between the demographic composition of the population and the evolutionary success of one of the traits is an example of the eco-evolutionary feedback present in this system.

Although the selective and ecological forces at play in the lab and in field conditions may differ significantly, our results predict that both forces have played a major role in the evolution of the rhizobium-legume mutualism by favoring the fixation of emerging N_2-fixing sub-populations among uncooperative symbiotic populations as well as their evolutionary maintenance. Yet the uncooperative population does not become extinct within nodules, likely because sanctions mainly target bacteroids of the nitrogen fixation zone. Releasing non-fixing bacteria may allow progenitors to meet appropriate hosts or to evolve new symbiotic traits. This loose selection process helps maintain genetically diverse rhizobial communities in the soil and shape the ecology and evolution of rhizobia. More generally, acknowledging the existence of non-cooperators as an integral component of the ecological and evolutionary dynamics of mutualistic interactions may provide a better understanding of the long-term persistence of bacterial lineages (*Heath and Tiffin, 2009*; *Heath and Stinchcombe, 2014*; *Tarnita, 2017*; *Fiegna et al., 2006*; *Hammerschmidt et al., 2014*).

An emerging trend in fundamental and applied plant microbiology is to select upon microbes indirectly through the host (*Mueller and Sachs, 2015*). This engineering approach, called host-mediated selection, involves selection of microbial traits that are not selectable in vitro. Modelling the eco-evolutionary scenarios provides predictions to guide experimental evolution studies aiming at designing beneficial microbes (*Marchetti et al., 2010*; *Marchetti et al., 2017*) and microbiomes (*Mueller and Sachs, 2015*; *Johns et al., 2016*).

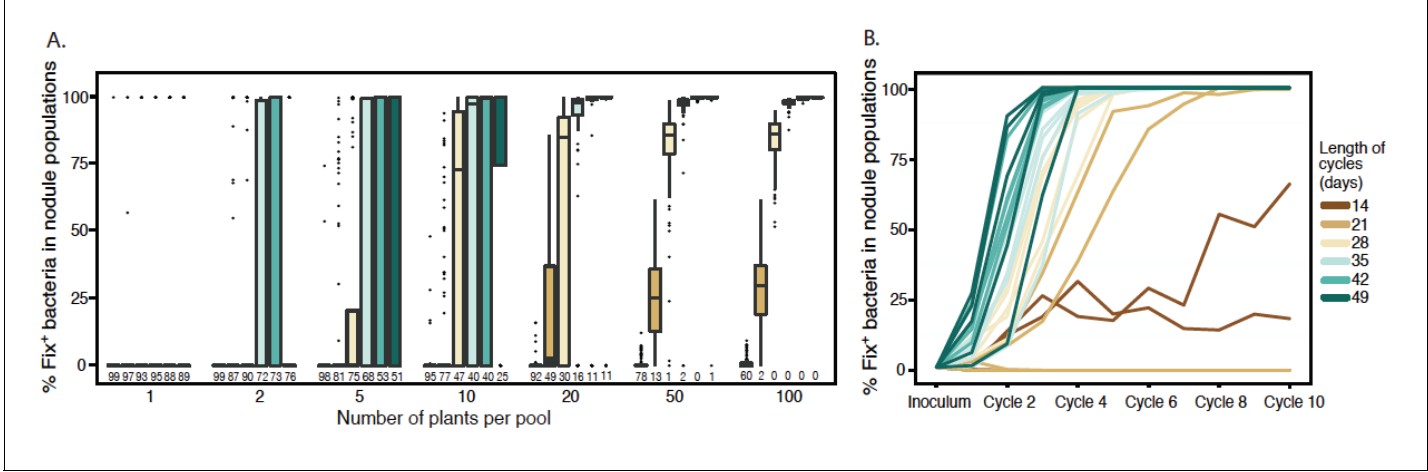

**Figure 9.** Effect of cycle length and plant numbers on the predicted distributions of Fix$^+$ population sizes. Model simulations were performed with an initial proportion of 1% Fix$^+$ in the bacterial population inoculated to a pool of plants. The length of each cycle and the number of plants per pool varied as indicated in the legend. (**A**) Final proportion of Fix$^+$ clones after four cycles (*Figure 9—source data 1*). Boxplots represent the distribution of the final proportion of Fix$^+$ clones from 100 simulations. The length of inoculation cycles ranged from 14 to 49 days and the number of plants per pool from 1 to 100. Numbers underneath each boxplot indicate the number of replicate simulations where Fix$^+$ sub-populations became extinct after four cycles. (**B**) Increase in the proportion of Fix$^+$ clones along 10 inoculation cycles of 14, 21, 28, 35, 42 or 49 days (*Figure 9—source data 2*). The number of plants per pool was 20. Representative trajectories of 5 replicate pools are shown in each case.

DOI: https://doi.org/10.7554/eLife.28683.022

The following source data and figure supplement are available for figure 9:

**Source data 1.** Simulation data for the final proportion of Fix$^+$ bacteria after four inoculation cycles.
DOI: https://doi.org/10.7554/eLife.28683.024

**Source data 2.** Simulation data for the increase in proportion of Fix$^+$ bacteria along 10 cycles.
DOI: https://doi.org/10.7554/eLife.28683.025

**Source data 3.** Simulation data for the effect of cycle length and plant number on the Fix$^+$ population sizes after four cycles.
DOI: https://doi.org/10.7554/eLife.28683.026

**Figure supplement 1.** Effect of cycle length and plant numbers on the predicted distribution of Fix$^+$ population sizes.
DOI: https://doi.org/10.7554/eLife.28683.023

# Materials and methods

## Bacterial strains and growth conditions

Strains and plasmids used in this study are listed in *Table 2*.

C. taiwanensis strains were grown at 28°C on TY medium supplemented with 6 mM CaCl$_2$ and 200 μg/ml streptomycin. *E. coli* strains were grown at 37°C on LB medium and antibiotics were used at the following concentrations: kanamycin 25 μg/ml, trimethoprim 100 μg/ml, tetracycline 10 μg/ml. For in vitro competition experiments, strains were pre-cultured in TY medium, mixed in equal proportion then co-inoculated to a 100 ml culture in TY medium. Bacteria were plated every 2 hr during the exponential phase, at the entry of stationary phase and 15 hr after the entry into the stationary phase. Plated bacteria were grown for 48 hr at 28°C then green and red bacteria were counted using a fluorescence stereo zoom microscope (Axiozoom V16, Zeiss).

## Mutant construction

Mutant and labeled strains of *C. taiwanensis* were constructed using the mutagenesis system developed by Flannagan et al. (*Flannagan et al., 2008*) involving the suicide plasmid pGPI-*Sce*I carrying an I-*Sce*I recognition site and the pDAI-*Sce*I replicative plasmid expressing the I-*Sce*I nuclease. To construct the unmarked *C. taiwanensis nifH* mutant, regions upstream and downstream *nifH* were amplified with the oCBM1821-oCBM2362 and oCBM1822-oCBM2363 primer pairs using GoTaq DNA polymerase (Promega). The two PCR products were digested with *Xba*I-*Bam*HI and *Bam*HI-*Eco*RI respectively and cloned into the pGPI-*Sce*I plasmid digested by *Xba*I and *Eco*RI. Ligation

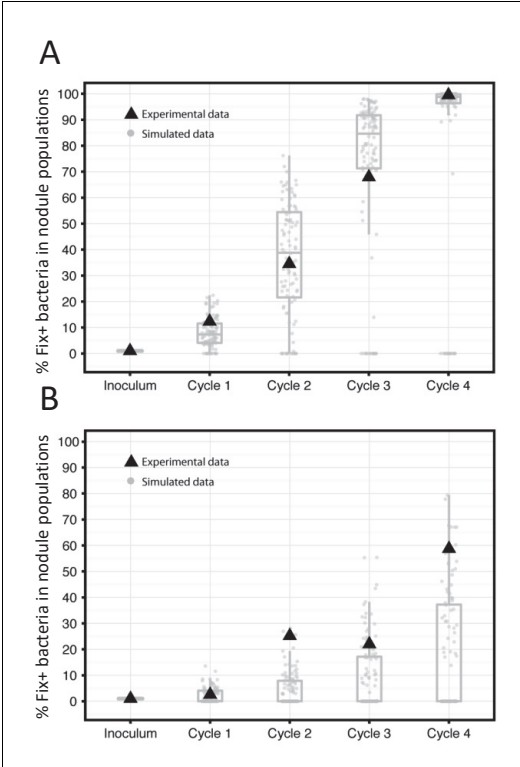

**Figure 10.** Frequency of Fix$^+$ bacteria over 4 cycles of 35 (**A**) or 21 (**B**) days: simulations and experimental validation. The proportion of Fix$^+$ clones over four inoculation cycles was simulated and measured experimentally. Simulations and experiment were performed with an initial proportion of Fix$^+$ clones of 1% and pools of 20 plants. Experiments were performed with an inoculum of $5 \times 10^3$ Fix$^+$/$5 \times 10^5$ Fix$^-$ *C. taiwanensis* per plant. The results from 100 replicate simulations are shown as grey dots and boxplots (*Figure 10—source data 1*). Experimental data are shown as black triangles (*Figure 10—source data 2*).

DOI: https://doi.org/10.7554/eLife.28683.027

The following source data is available for figure 10:

**Source data 1.** Simulation data for the frequency of Fix$^+$ bacteria over 4 cycles of 35 or 21 days.
DOI: https://doi.org/10.7554/eLife.28683.028

**Source data 2.** Experimental data for the frequency of Fix$^+$ and Fix$^-$ bacteria over 4 cycles of 35 or 21 days.
DOI: https://doi.org/10.7554/eLife.28683.029

products were transformed into a DH5α λpir *E. coli* strain. The resulting plasmid was transferred into *C. taiwanensis* CBM832 by triparental mating using pRK2013 as helper plasmid. Transconjugants that have integrated the plasmid by single crossing over were selected on streptomycin and trimethoprim and verified by PCR using the oCBM1824-oCBM2363 and oCBM1825-oCBM2362 primer pairs. Then we introduced the pDAI-*Sce*I replicative plasmid into these strains by conjugation and selection on tetracyclin. Expression of the I-*Sce*I nuclease causes a double strand break into the inserted plasmid and promotes DNA recombination. Mutants deleted in *nifH* were screened by trimethoprim sensitivity and verified by PCR using the oCBM1824-oCBM1825 pair of primers. Mutants were then cultivated on unselective TY medium. Tetracycline sensitive colonies which have lost the pDAI-*Sce*I plasmid were selected.

The P*ps*-GFP and P*ps*-mCherry fusions were inserted into the wild-type and *nifH* mutant of *C. taiwanensis* at the same chromosomal locus, i. e. in the intergenic region between the *glmS* and RALTA_A0206 genes using the same pGPI-*Sce*I/pDAI-*Sce*I mutagenesis system. Flanking regions of the insertion site were amplified by PCR using the Phusion DNA polymerase (ThermoFisher Scientific) and the oCBM2619-oCBM2620 and oCBM2621-oCBM2622 primer pairs. PCR products were digested by XbaI and Acc65I or Acc65I and EcoRI respectively and cloned into the pGPI-*Sce*I plasmid digested by XbaI and EcoRI. The two fusions P*ps*-GFP and P*ps*-mCherry were obtained by digesting the pRCK-P*ps*-GFP and pRCK-P*ps*-mCherry by AvrII and SpeI and cloned into the pGPI-*Sce*I carrying the intergenic region *glmS*-RALTA_A0206 digested by the same enzymes. The resulting pCBM161 and pCBM162 were first transformed into a DH5α λpir *E. coli* strain then transferred into *C. taiwanensis* by triparental mating with the pRK2013 helper plasmid. Integration of the fusions by double crossing over was carried out using the pDAI-*Sce*I plasmid as described above. CBM2700 (Fix$^+$, GFP) and CBM2707 (Fix$^-$, mCherry) had the same plating efficiency in in vitro competition experiments, indicating that these genetic modifications did not noticeably affect bacterial growth rate.

Oligonucleotide sequences used for genetic constructions are provided in *Supplementary file 1*.

## Plant tests

*Mimosa pudica* seeds were of Australian origin (B and T World Seed, Paguignan, France) and were sterilized as described (*Chen et al., 2003*). Seedlings were cultivated in Gibson tubes (2 *M. pudica* plantlets/tube) as previously described (*Marchetti et al., 2014*). To increase the frequency of co-infection, plants were grown on 12 cm$^2$ plates (three plants per plate) containing slanting nitrogen-

**Table 2.** Strains and plasmids used in this study

| Bacterium | Strain | Relevant characteristics | Reference/source |
|---|---|---|---|
| E. coli | DH5α | F recA lacZM15 | Bethesda research laboratory |
| | DH5α λpir | F recA lacZM15 λpir | HP Schweizer |
| C. taiwanensis | LMG19424 | Wild-type strain isolated from Mimosa pudica in Taiwan | (Chen et al., 2001) |
| | CBM832 | LMG19424 derivative resistant to Streptomycin, Str$^R$ | M. Hynes |
| | CBM2568 | CBM832 deleted in nifH, Str$^R$ | This study |
| | CBM2700 | CBM832 carrying a Pps-GFP fusion downstream glmS, Str$^R$ | This study |
| | CBM2701 | CBM2568 carrying a Pps-GFP fusion downstream glmS, Str$^R$ | This study |
| | CBM2707 | CBM2568 carrying a Pps-mCherry fusion downstream glmS, Str$^R$ | This study |
| | CBM2708 | CBM832 carrying a Pps-mCherry fusion downstream glmS, Str$^R$ | This study |
| Plasmids | Name | Relevant characteristics | Reference/source |
| | pGPI-SceI | ori$_{R6K}$, mob$^+$, carries a I-SceI site, Tri$^R$ | (Flannagan et al., 2008) |
| | pDAI-SceI | ori$_{pBBR1}$, mob$^+$, carries the I-SceI gene, Tet$^R$ | (Flannagan et al., 2008) |
| | pRCK-Pps-GFP | Plasmid carrying the psbA promoter region fused to GFP, Kan$^R$ | M. Valls |
| | pRCK-Pps-mCherry | Plasmid carrying the psbA promoter region fused to mCherry, Kan$^R$ | M. Valls |
| | pCBM156 | pGPI-SceI carrying the nifH 5' and 3' regions, Tri$^R$ | This study |
| | pCBM161 | pGPI-SceI carrying the glmS-Ralta_A0206 intergenic region interrupted by a Pps-GFP fusion, Tri$^R$ | This study |
| | pCBM162 | pGPI-SceI carrying the glmS-Ralta_A0206 intergenic region interrupted by a Pps-mCherry fusion, Tri$^R$ | This study |
| | pRK2013 | Helper plasmid, Kan$^R$ | (Figurski and Helinski, 1979) |

Str, spreptomycin; Tri, trimethoprim; Tet, tetracycline; Kan, kanamycin.

DOI: https://doi.org/10.7554/eLife.28683.030

free Fahraeus agar medium for 3 days at 28°C. Roots were covered with a sterile, gas-permeable, and transparent plastic film (BioFolie 25; Sartorius AG, Vivascience, Bedminster, NJ, U.S.A.). For single-strain inoculation experiments, each plant in Gibson tubes was inoculated with $5.10^5$ bacteria either CBM832 (wild-type) or its isogenic nifH mutant, CBM2568. For co-inoculation experiments in Gibson tubes, plants were inoculated with the two isogenic strains CBM2700 (wild-type, GFP labeled) and CBM2707 (nifH, mCherry labeled) at ratio 1/1 ($5.10^5$ bacteria of each strain per plant) or 1/100 ($5.10^3$ bacteria of CBM2700 and $5.10^5$ bacteria of CBM2707 per plant). For co-inoculation experiments in plates, plants were inoculated with $10^{10}$ bacteria of each strain per plant.

To measure the number of nodule bacteria over time, all nodules from 5 to 10 individual plants, except very small nodules, were individually collected with at least 2 mm of root left on both sides of nodules and treated at each time point. We did not collect very small nodules since there was a risk that the sterilization agents penetrate these nodules. In the same line we did not collect nodules before 14 dpi since most nodules were very small at that stage. Nodules were surface sterilized for 15 min in a 2.5% sodium hypochlorite solution, rinsed with water and crushed. Each nodule crush was diluted and plated using an easy spiral automatic plater (Interscience). Colonies were counted after 2 day-incubation at 28°C, under a fluorescence stereo zoom microscope (Axiozoom V16, Zeiss) when appropriate.

For nodulation kinetics, the number of nodules formed on 20 plants grown in Gibson tubes was counted daily for 6 weeks.

For serial inoculation-nodulation cycles on M. pudica plants, 10 Gibson tubes of plants were inoculated with CBM2700 and CBM2707 in 1/100 ratio as described above. 35 days after inoculation, all nodules were collected, surface-sterilized and crushed together. The nodule crush was used to inoculate a new set of 10 tubes of plants with 50 µl of a 1/10 dilution of the nodule crush per plant. At

each cycle, dilutions of the nodule crush were spread on plates, incubated 2 days at 28°C and colonies were counted under a fluorescence stereo zoom microscope.

## Cytological analyses

The viability of nodule bacteria was estimated using propidium iodide staining at a concentration of 20 mM in DMSO (Molecular Probes, Fisher scientific, Oregon) on 55/58 μm nodule sections. For each experiment, a dozen nodules were individually analyzed at 14, 16, 17, 21, 28 and 35 dpi. For electron microscopy analysis, nodules were fixed in glutaraldehyde (2.5% in phosphate buffer 0.1 M [pH 7.4]), osmium treated, dehydrated in an alcohol series, and embedded in Epon 812. Semithin nodule sections were observed by brightfield microscopy after staining in 0.1% aqueous toluidine blue solution and observed under a Zeiss Axiophot light microscope. Ultrathin sections were stained with uranyl acetate and observed with a TEM Hitachi HT7700.

## Acetylene reduction assays

*M. pudica* plants were inoculated with the wild-type strain of *C. taiwanensis* CBM832. At different time points, plants were removed from the culture Gibson tube and placed in an airtight tube and incubated with 1 ml of acetylene for 4 hr. 100 μl of gas were then injected into a gas chromatograph (Agilent GC7820). The area of the ethylene peak was measured and compared to an ethylene standard of known concentration. Ethylene background was estimated by analyzing empty tubes incubated with the same amount of acetylene.

## Mathematical model and simulations

The model aimed at simulating nodulation dynamics during single or repeated inoculation-nodulation cycles. First we parameterized the population dynamics during the symbiosis process. Then we simulated repeated nodulation cycles varying the following parameters: (i) the Fix$^+$/Fix$^-$ ratio in the initial inoculum, (ii) the number of inoculated plants, and (iii) the cycle length. The model ran on a pool of plants (of given, variable size) from which nodules were collected and mixed together after each inoculation cycle. For each time-step (1 day) after inoculation, the number of new nodules formed on each plant was randomly drawn from a Poisson distribution of parameter $\lambda(t, nod^+_t)$, which is itself a function of time $t$ and of the number of nodules already present on the plant $nod^+_t$ at time $t$. The maximal number of nodules that could potentially be formed per day per plant was set to $\lambda_{max}$. Changing the value of parameter $\lambda$ depending on the number of Fix$^+$nodules already present on the plant simulated the autoregulation of nodulation process; this was done by subtracting the factor $a_1 \times nod^+_t$ from $\lambda_{max}$. Lastly, to allow for some 'aging' process that would decrease the rate of nodulation with time (even for plants inoculated only with Fix$^-$ bacteria), we incorporated a time-decay coefficient: $a_2 \times (t- a_3)$, meaning that a reduction in the rate of nodulation occurred at a rate $a_2$ when $t > a_3$. This time-decay factor was set to 0 when $t < a_3$. Therefore, the parameter of the Poisson distribution controlling the rate at which new nodules are formed was given by: $\lambda(t, nod^+_t) = \lambda_{max} - a_1 \times nod^+_t$ for $t < a_3$ and by: $\lambda(t, nod^+_t) = \lambda_{max} - a_1 \times nod^+_t - a_2 \times (t- a_3)$ for $t > a_3$. Since nodules are persistent once formed, we further set: $\lambda(t, nod^+_t) \geq 0$. Experimental evidence indicated that the number of inoculated bacteria did not affect nodulation kinetics as long as the total inoculum remains above $10^3$ bacteria per plant. These conditions were met in all experiments described in this work. Therefore, we did not explicitly take inoculum size into consideration in the simulations, and restricted the applicability of our model to cases where inoculum was above this threshold value.

The second module of the model dealt with bacterial multiplication within plant nodules. Within each nodule we assumed a logistic growth model for the bacteria given by: $X(t + 1) = (r-c -su_{ds}(t)) \times X(t) \times (1-X(t)/K)$, where $r$ was the growth rate, $c$ the net fitness cost of nitrogen fixation in Fix$^+$ bacteria, $su_{ds}(t)$ the additional plant sanctions against Fix$^-$ bacteria occurring in the later phase of the interaction, $X(t)$ the bacterial population at time $t$ and $K$ the nodule carrying capacity. In our simulations, we set $c = 0$ since we experimentally did not detect any difference in the populations of Fix$^-$ or Fix$^+$ nodule bacteria at 14 dpi. We emphasize that a net fitness cost of 0 does not necessarily imply that nitrogen fixation does not impose a metabolic burden on the bacteria (referred to as 'metabolic cost' in the results section). Instead, this burden, if significant during the early steps of the interaction, may be compensated for by plant control mechanisms acting at a basal level. Beyond this time

point, additional plant sanctions (possibly including partner fidelity-feedback) were given by $su_{ds}(t)$, taking the value $s$ of plant sanctions indicated in *Table 1* as long as the age of the nodule was higher than $ds$ days (denoted by the step function $u_{ds}(t)=0$ if $t < ds$ or $u_{ds}(t)=1$ if $t > ds$).

Parameters values were estimated by computing the minimal root mean square error (RMSE) of experimental data (nodulation kinetics and bacterial multiplication within nodules) versus model outputs calculated for a range of parameter values. Parameter values selected to minimize RMSE are indicated in *Table 1*. Simulations were implemented in R (*R Core Team, 2014*) and code is available in the *Source code file 1*.

## Acknowledgements

We are grateful to Jacques Batut, Peter Young and Erik Hom for helpful comments on the manuscript. This work was supported by funds from the French National Research Agency (ANR-12-ADAP-0014-01 and ANR-16-CE20-0011-01), and the French Laboratory of Excellence project 'TULIP' (ANR-10-LABX-41; ANR-11-IDEX-0002-02). BD was supported by an INRA-Région Occitanie fellowship. CSG acknowledges support from the Max Planck Society.

## Additional information

### Funding

| Funder | Grant reference number | Author |
|---|---|---|
| Agence Nationale de la Recherche | ANR-12-ADAP-0014-01 | Marta Marchetti<br>Catherine Masson-Boivin<br>Delphine Capela |
| Institut National de la Recherche Agronomique | | Benoit Daubech |
| Max-Planck-Gesellschaft | | Chaitanya S Gokhale |
| Agence Nationale de la Recherche | ANR-16-CE20-0011-01 | Marta Marchetti<br>Catherine Masson-Boivin<br>Delphine Capela |
| Agence Nationale de la Recherche | ANR-10-LABX-41 | Benoit Daubech<br>Marta Marchetti<br>Cécile Pouzet<br>Marie-Christine Auriac<br>Catherine Masson-Boivin<br>Delphine Capela |
| Agence Nationale de la Recherche | ANR-11-IDEX-0002-02 | Benoit Daubech<br>Marta Marchetti<br>Cécile Pouzet<br>Marie-Christine Auriac<br>Catherine Masson-Boivin<br>Delphine Capela |

The funders had no role in study design, data collection and interpretation, or the decision to submit the work for publication.

### Author contributions

Benoit Daubech, Marta Marchetti, Cécile Pouzet, Marie-Christine Auriac, Investigation, Methodology; Philippe Remigi, Conceptualization, Formal analysis, Investigation, Methodology, Writing—original draft, Writing—review and editing; Ginaini Doin de Moura, Investigation; Chaitanya S Gokhale, Conceptualization, Supervision, Methodology, Writing—review and editing; Catherine Masson-Boivin, Conceptualization, Supervision, Funding acquisition, Methodology, Writing—original draft, Writing—review and editing; Delphine Capela, Conceptualization, Data curation, Formal analysis, Supervision, Investigation, Methodology, Writing—original draft

## Author ORCIDs

Philippe Remigi http://orcid.org/0000-0001-9023-3788
Chaitanya S Gokhale http://orcid.org/0000-0002-5749-3665
Catherine Masson-Boivin http://orcid.org/0000-0002-3506-3808

## Decision letter and Author response

Decision letter https://doi.org/10.7554/eLife.28683.034
Author response https://doi.org/10.7554/eLife.28683.035

## Additional files

### Supplementary files

• Supplementary file 1. Primers used in this study.
DOI: https://doi.org/10.7554/eLife.28683.031

• Source code file 1. R code used for simulations.
DOI: https://doi.org/10.7554/eLife.28683.032

• Transparent reporting form
DOI: https://doi.org/10.7554/eLife.28683.033

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
