## [Decision Letter]

Thank you for submitting your article "Spatio-temporal control of mutualism in legumes helps spread symbiotic nitrogen fixation" for consideration by *eLife*. Your article has been reviewed by three peer reviewers, one of whom, Wenying Shou, is a member of our Board of Reviewing Editors. Ian Baldwin oversaw the process as the Senior Editor. The following individuals involved in review of your submission have agreed to reveal their identity: Maren Friesen (Reviewer #2) and R. Ford Denison (Reviewer #3).

The reviewers have discussed the reviews with one another and the Reviewing Editor has drafted this decision to help you prepare a revised submission.

Summary:

Baubech et al. examined how legumes deal with rhizobium that does not fix nitrogen. We find parts of the paper very interesting. However, we have problems with your data interpretation, especially when you regard nitrogen fixation as cost-free in your model and in your text. Seems that your results are consistent with partner-fidelity feedback control of non-fixing bacteria (i.e. host cells with fix^-^ mutants don't do well, and thus fix^-^ mutant do not do well). We encourage you to rethink and rewrite.

Reviewer #1:

My feelings for this paper are mixed. I like how the authors showed that nitrogen-starved plants had increased nodules over time (presumably until plant's nitrogen need is met). In a single nodule, fix^+^ and fix^-^ nodule cells are spatially partitioned, and fix^+^ increases its frequency over fix^-^. Fix^+^ can increase in frequency during several rounds of inoculation/growth, and this increase is faster in longer cycle (as predicted by theoretical work decades ago). However, the paper made quite problematic assertions, which rendered the model and the entire paper problematic.

Subsection “Evidence for a spatial and temporal control of mutualism in Mimosa nodules”: "Since non-fixers do not proliferate better than fixers (Figure 1), they are not cheaters in our system following a fitness-based definition of the term "cheater". This statement (and the thinking behind it, including assuming that nitrogen-fixation has zero cost in modeling) is problematic. You are dealing with a spatially-structured environment. Given the spatial segregation of fix^+^ and fix^-^ cells to different nodule cells, partner fidelity feedback (see Bull and other's work, which is further extended in https://elifesciences.org/articles/10106) cannot be ruled out. In fact, partner fidelity feedback is likely to operate. In other words, plant cells near fix^-^ bacteria do poorly, and thus fix^-^ bacteria do poorly as well. If you stand behind your claim that N2-fixation is cost free, then you must back up this extraordinary claim with exceptionally careful experiments. That is, if you were to mix the two types together so that they are well-mixed in a plant-like environment (which is probably difficult to do and not done here), fix^-^ cells are more fit or less fit than fix^+^ cells?

Figure 1, Figure 3 understand fix^+^ and fix^-^ bacteria, but I am confused about fix^+^ and fix^-^ nodule. What is the definition?

Figure 2/C: I am not convinced: there are no data points before dpi 14, and so growth rate cannot be credibly estimated.

Reviewer #2:

This is a significant contribution to the field of rhizobia-legume symbiosis, demonstrating nicely that non-fixing rhizobia are penalized both at the whole nodule level as well as within nodules – a matter of much debate! – and going farther to considering the time-course of this process. The modeling is a nice addition, though not necessarily required for this to be a strong paper. The multigenerational experiments seal the deal, demonstrating that the phenomenon measured within a generation translates to the expected shifts in allele frequencies over time.

I have only two concerns:

First, as this is something I worry about a lot in my own work, to what extent does the surface sterilization process penetrate and kill rhizobia within Mimosa nodules? This could have dramatic effects on the rhizobial populations observed in single-strain nodules that vary in size, which typically correlates with fixation status.

Second, the authors claim in the text that the model fits their experimental data very well, but do not justify this statistically. In fact, looking at Figure 2 and Figure 10, the model doesn't appear to fit all that well for particular timepoints and there doesn't seem to be all that much power in rejecting the model, if this was in fact the intent. Some consideration of what the purpose of the model is would be warranted – I think it is a nice conceptual addition that shows qualitatively that cycle length and plant population size impact the evolution of fix^+^ from low frequency, but that the quantitative fit is being overstated.

Reviewer #3:

This paper presents some important data, although some of the interpretation is questionable. The important results are:

1) Host sanctions against a nonfixing rhizobial strain were much more severe than previously reported for other host species (Figure 1). In the real world, however, nonfixing nodulating strains like that used in the experiments are rare, relative to less-beneficial strains that fix some nitrogen. Furthermore, there was no evidence that the resources the nifH strain spared by not fixing nitrogen are automatically invested in rhizobial fitness. Therefore, we cannot conclude (subsection “Evidence for a spatial and temporal control of mutualism in Mimosa nodules”) that "low quality rhizobial partners rarely benefit from low investment in mutualism." (If there were a consensus on this, as claimed, this paper would merit publication only if it challenged that consensus.) To test that hypothesis would require comparing two nitrogen-fixing strains, with one of them providing less benefit via a mechanism that would plausibly enhance its own fitness. Examples include strains that (once inside nodules) block nodulation by other rhizobia (Tatsukami, 2016) or strains that divert more resources from N2 fixation to their own reproduction. Nonfixing mutants would not qualify.

2) Sanctions were apparently imposed on a nonfixing strain within mixed nodules (Figure 1). This is arguably the most-important result, so should be featured in the Abstract, along with the "clear sectoring" (subsection “Evidence for a spatial and temporal control of mutualism in Mimosa nodules") that is presumably key to within-nodule discrimination by the host.

3) Figure 3 shows no partner choice (host-imposed discrimination during the infection process). This is consistent with past work comparing isogenic strains, but the authors should recognize the possibility of adaptive partner choice by some host plants among some real-world strains whose signals are correlated with their relative benefits.

Given these empirical results, it's obvious that fix^+^ rhizobia could invade a fix^-^ population under real-world conditions. I don't really see the point to modeling cases with only 20 plants or where plants only live 14 days after rhizobial infection, as either of those would lead to extinction of the host. If the authors want to explore "ecological factors", as claimed, they should explore some more-realistic factors like temperature or soil nitrogen. What about competition between a strain that fixes half as much nitrogen as another, diverting the resources saved to its own reproduction? That's more realistic than a nonfixing strain. Their relative fitness would depend on how the host's threshold for imposing sanctions and on the extent to which resources diverted from nitrogen fixation actually enhance rhizobial fitness. Until we have those data, modeling is almost pointless. If space in the journal is not an issue, they could keep the modeling work, but the Abstract should be rewritten to be less vague and to highlight the empirical results. One sentence in the Abstract on modeling would be plenty.

[Editors' note: further revisions were requested prior to acceptance, as described below.]

Thank you for submitting your revised article "Spatio-temporal control of mutualism in legumes helps spread symbiotic nitrogen fixation" for consideration by *eLife*. Your revision has been evaluated by Ian Baldwin in consultation with a Reviewing Editor.

Thank you for your revision, but we are returning the revision without further review, because we don't think that you have adequately addressed the concerns about the "costs" that play a central role in your modeling effort (or we were not sufficiently clear in our previous decision letter). In your model, the "cost of nitrogen fixation" being zero strains credulity unless you are very specific about what that these costs mean.

Hence, we think that you need to be very specific about how these costs are described; otherwise the modelling effort could be criticized as being a tautology. If we understand your arguments correctly, the "post-PFF/post-sanctioning costs" need to be very small or even negative (benefit of cooperation) for cooperation to persist. Hence the text should be revised to say something like "the cost of cooperation is difficult to determine because such determination requires a well-mixed environment to be free of PFF. Regardless, this cost must have been overcome by PFF and/or sanctioning mechanisms. The cost we refer to is post-PFF/post-sanctioning cost of cooperation, which should be zero or negative, otherwise the mutualisms would have failed".

We hope that you can address this concern in short order and we look forward to subjecting your revision to a full re-review.

[Editors' note: further revisions were requested prior to acceptance, as described below.]

Thank you for submitting your article "Spatio-temporal control of mutualism in legumes helps spread symbiotic nitrogen fixation" for consideration by *eLife*. Your article has been reviewed by three peer reviewers, one of whom, Wenying Shou is a member of our Board of Reviewing Editors and the evaluation has been overseen by Ian Baldwin as the Senior Editor. The following individuals involved in review of your submission have agreed to reveal their identity: Maren Friesen (Reviewer #2); R. Ford Denison (Reviewer #3).

The reviewers have discussed the reviews with one another and the Reviewing Editor has drafted this decision to help you prepare a revised submission.

Summary:

Reviewers feel that you have made improvements to the manuscript. There are points that still need to be addressed.

Essential revisions:

1) Since their most interesting result was apparent sanctions within mixed nodules, they should cite this just-published paper: http://www.amjbot.org/citmgr?gca=amjbot;ajb.1700165v1

2) You will need to explain/contrast sanctions and PFF clearly in Intro to give readers sufficient background.

3) A clarification of "cost".

Reviewer #2

Cost definitions are key. In the model, the cost is necessarily a fitness cost because it is translating fixation rate into population growth rate (i.e., per capita fitness). This is, as articulated in the response to reviewers’ letter, completely distinct from the actual metabolic cost of nitrogen fixation in terms of ATP and reducing power. I don't see the point is having separate terms for fitness cost of fixation (c) and then the fitness effect of sanctions (s1) since these are tied together in this system anyways. It would be clearer to just have a "net fitness effect" of fixation ability that could just be zero, because the growth rate of the fix^+^ and fix^-^ is the same early in the interaction. The distinction between early phase vs late-phase seems arbitrary.

Related to this, when do Mimosa nodules really start fixing nitrogen? Prior to this point one wouldn't expect there to be a difference between fix^+^ and fix^-^ since the trait isn't expressed. In Figure 7, it isn't clear whether the first time-point (7d) is significantly different from zero – this would be worth presenting.

Subsection “Evidence for a spatial and temporal control of mutualism in Mimosa nodules”: Should specify here that you mean the fitness cost; also, the double negative "preventing mutualism to fail", is confusing and should be revised to "enabling mutualism to spread".

It should be clarified which cells are contributing to the population when nodules are crushed and plated – are these a mix of former bacteroids (the ~20% that are supposedly culturable–so what is happening to the other 80%) and cells from the infection threads? If most of the cells come from infection threads, this could dilute the effect of plant control unless the plant is able to also regulate cells in ITs on the basis of the neighboring bacteroids… Please clarify what you think is happening in this system, as this could guide the next steps in dissecting this phenomenon.

Subsection “Eco-evolutionary dynamics of N2-fixers and non-fixers through serial nodulation cycles”: Should specify that you are qualitatively testing the model.

Discussion section: degeneration

Discussion section: dashes don't match

Methods: tetracyclin –> tetracycline

Reviewer #3:

I continue to favor publishing the empirical results, while remaining concerned about the distinction between metabolic cost and opportunity cost.

If, as the authors state that in their response, they "have no evidence that nifH mutants can invest more resources in their own fitness," then their results are only relevant to cases where there is no opportunity cost (in terms of potential rhizobial fitness, not just metabolism) to fixing nitrogen. For the Abstract to accurately describe their results, it would have to read something like "rare fixers will invade a population of nonfixing bacteria that lack mechanisms to divert resources from nitrogen fixation to their own fitness…" There's plenty of fluff in the Abstract that could be cut to include this key qualification.

Similarly, where they speculate that "the metabolic cost paid by bacteria to fix nitrogen is too low to be detected" the issue isn't metabolic cost but opportunity cost. Even if metabolic cost is very high, the resources are supplied by the plant. And, if diverting some of those resources to rhizobial reproduction isn't possible, then metabolic costs are irrelevant to rhizobial fitness. That is apparently the case for their nifH mutant. The lack of difference at 14 dpi would only be "surprising" if nifH mutants were able to divert resources to their own reproduction, prior to the imposition of sanctions.

The real question, though, is whether strains that fix less nitrogen and can divert resources saved to their own reproduction can out-compete strains that fix more nitrogen. If we accept the importance of this question, then the empirical results merit publication, because the sanctions shown in Figure 1 are severe enough to outweigh any likely fitness benefits of diverting resources from nitrogen fixation. They don't have any data on the fitness effects of fixing less nitrogen, rather than none – this might not trigger sanctions – but that's not a reason not to publish. It's just a reason not to over-generalize in Abstract and main text. For example, in the Discussion section "in our system" is too vague, especially when coupled with generalizations about "the absence of a social dilemma." That would be a good place to point out that results could be different for "cheaters" (as opposed to "losers"), that is, strains that fix some nitrogen, but divert more resources to their own reproduction than other strains do.

I accept the argument that a model that only applies to small experimental evolution studies (because of low plant numbers) is potentially useful, but the reference to "ecological factors" in the Abstract promises too much.

---

## [Author Response]

Summary:Baubech et al. examined how legumes deal with rhizobium that does not fix nitrogen. We find parts of the paper very interesting. However, we have problems with your data interpretation, especially when you regard nitrogen fixation as cost-free in your model and in your text. Seems that your results are consistent with partner-fidelity feedback control of non-fixing bacteria (i.e. host cells with fix^-^ mutants don't do well, and thus fix^-^ mutant do not do well). We encourage you to rethink and rewrite.Reviewer #1:My feelings for this paper are mixed. I like how the authors showed that nitrogen-starved plants had increased nodules over time (presumably until plant's nitrogen need is met). In a single nodule, fix^+^ and fix^-^ nodule cells are spatially partitioned, and fix^+^ increases its frequency over fix^-^. Fix^+^ can increase in frequency during several rounds of inoculation/growth, and this increase is faster in longer cycle (as predicted by theoretical work decades ago). However, the paper made quite problematic assertions, which rendered the model and the entire paper problematic.Subsection “Evidence for a spatial and temporal control of mutualism in Mimosa nodules”: "Since non-fixers do not proliferate better than fixers (Figure 1), they are not cheaters in our system following a fitness-based definition of the term "cheater". This statement (and the thinking behind it, including assuming that nitrogen-fixation has zero cost in modeling) is problematic. You are dealing with a spatially-structured environment. Given the spatial segregation of fix^+^ and fix^-^ cells to different nodule cells, partner fidelity feedback (see Bull and other's work, which is further extended in https://elifesciences.org/articles/10106) cannot be ruled out. In fact, partner fidelity feedback is likely to operate. In other words, plant cells near fix^-^ bacteria do poorly, and thus fix^-^ bacteria do poorly as well. If you stand behind your claim that N2-fixation is cost free, then you must back up this extraordinary claim with exceptionally careful experiments. That is, if you were to mix the two types together so that they are well-mixed in a plant-like environment (which is probably difficult to do and not done here), fix^-^ cells are more fit or less fit than fix^+^ cells?

First, we need to clarify our use of the terms ‘cheater’ and ‘cost of nitrogen fixation’.

We employed the term ‘cheater’ in the sense defined by Jones et al., 2015, as we believe that this definition is helpful to think about the evolutionary dynamics of rhizobial populations. This definition states that “cheating must *increase the fitness of the actor above average fitness in the population* and *decrease the fitness of the partner below average fitness in the partner population” (Jones, 2015).* This definition therefore considers ‘cheating outcome’ rather than ‘cheating actions’. *Although we did not measure plant fitness –our analysis is therefore partial in that respect, w*e did not observe a better proliferation of non-fixers over fixers at analyzed time points, and thus non-fixers do not “behave” as “cheaters”. Nevertheless, we acknowledge our sentence could lead to confusion and have modified the sentence (see below).

In relation to the above definition, we used the term ‘cost of nitrogen fixation’ to quantify the net cost/benefit ratio of mutualism; once more we employed a fitness-based definition instead of focusing on the actual metabolic cost of nitrogen fixation. Because the Fix^+^/Fix^-^ fitness ratio is never in favor of the non-fixing strain, we set the cost of nitrogen fixation to 0 in the model (and used another parameter (‘sanctions’) to describe the observed fitness cost of non-fixing strains in older nodules). Several reasons could be invoked to explain the fact that we did not detect any fitness advantage for non-fixing strains: 1) there could be really no metabolic cost; this is however unlikely because 16 molecules of ATP are required to reduce 1 molecule of N2, 2) the metabolic cost could be too small or transient to be detected in our measures or 3) the metabolic cost could be compensated for by stabilizing mechanisms even during the early phases of the interaction. We have thus introduced these possibilities by changing the sentence “Since non-fixers do not proliferate better than fixers (Figure 1), they are not cheaters in our system” into “Surprisingly, non-fixers did not proliferate better than fixers even at 14 dpi (Figure 1) possibly because the metabolic cost of nitrogen fixation is too low to be detected in our experimental conditions, or because plant sanctions and the cost of nitrogen fixation equilibrate until sanctions become prominent” (subsection “Evidence for a spatial and temporal control of mutualism in Mimosa nodules”).

We have also included a more precise definition of ‘cost of nitrogen fixation’ in the description of the model: “We emphasize that our definition of ‘cost of nitrogen fixation’ therefore does not refer to the actual metabolic cost of this process. It rather refers to the net fitness effect acting on nitrogen-fixing strains, which may integrate the actual metabolic cost and potential plant sanction mechanisms that could be at work in the early phases of the interaction and thus compensate for the metabolic cost” (subsection “Mathematical model and simulations”).

In addition, we would like to note that *Cupriavidus taiwanensis*, like most rhizobia, fix nitrogen only in plant nodule cells that provide the appropriate environment for this: (i) a large bulk of plant photosynthetates is diverted to bacteroids solving the high ATP demand of the nitrogenase enzyme, (ii) the paradox between the extreme sensitivity of nitrogenase to oxygen and the strict (micro) aerobic status of rhizobia is solved in two ways: nodules exhibit an extremely low free oxygen atmosphere and nodule leghemoglobin – a form of the plant ubiquitous hemoglobin- facilitates O_2_ diffusion to bacteroids. Furthermore, gene expression in bacteroids is completely modified so that their metabolism is entirely dedicated to nitrogen fixation. Nitrogen fixation only occurs after bacteria have entered the root and massively invaded the nodule cells, i.e. in the fixation zone of the nodule but not in infection threads or in the infection zone (process both spatially and temporally regulated). It is thus impossible to evaluate the metabolic cost of nitrogen fixation in a plant-like environment. In free-living conditions, our isogenic Fix^+^ and Fix^-^ strains are equally fit (but they do not fix in these conditions).

Second, the concept of partner fidelity feedback was indeed not considered in our current manuscript. We thank the reviewer for the input as this prompted us to think in the direction of the exact ongoing mechanisms.

We do not know the mechanisms that drive the death of non-fixers and host cells in the fixation zone. PFF is an attractive model because it has the potential to stabilize mutualistic interactions without having to invoke additional, unknown mechanisms. The analogy with the yucca-pollinator detailed in Shou, 2015 is particularly interesting for us, since we scrutinized the legume-rhizobia interaction at the cellular level: PC at the nodule level could actually result from PFF at the cellular level.

However, several specificities of the legume-rhizobia interaction deserve further attention. Nodule cell and bacterial degeneration in Fix^–^ occupied nodules could result from a defense response from the plant side in response to a bacterial signal (or an absence of signal). A scenario of active plant defense response was recently discovered in Medicago truncatula (Wang et al., 2017; Yang et al., 2017) (plant peptides killing specific bacterial genotypes at the intracellular level) and seems to be consistent with PC through conditional response. Moreover, PFF assumes that plant cells (entities) are metabolically insulated. That is, a plant cell has a limited amount of resources, and it will die if bacteria use up all its nutrients, which we believe is unlikely. Nodules are carbon-sink organs, meaning that the plant devotes significant amounts of carbohydrates to the nodules, and nutrients are exchanged between infected and non-infected plant cells either through transporters and/or by diffusing in the apoplast (White et al., 2007). Non-infected cells play a key role in nutrient exchange between the nodule and the root (Godiard et al., 2011). Therefore, the relevance of the cellular entities for mutualism in the legume nodules is debatable.

Given the arguments explained in the above paragraph and the impossibility to test the spatial equivalence of the focal entities, we feel that we cannot decisively assess which mechanism operates in our system and that our results could be interpreted as either PC through conditional response or PFF. Moreover, another potential mechanism could be that the bacteroid status (complete modification of gene expression in the plant environmental conditions) is lethal to bacteria that do not produce a functional nitrogenase system or do not export nitrogen. Such mechanism could possibly be interpreted as PFF at the cellular level.

Since we do not know the molecular mechanisms responsible for the death of non-fixers, we prefer to be cautious and keep the expression “post-infection sanctions against non fixers” in the text (Discussion section), as commonly employed in the legume-rhizobia literature.

Yet, we have now clarified our definition of the term ‘sanction’, in a way that we believe is compatible with both models, and have mentioned the PFF possibility: “These sanctions could occur as defense responses and/or by decreasing nutrient supply to non-fixing bacteroids. Given that Fix^-^ and Fix^+^ bacteria are spatially segregated within nodules, the latter case could also result from the local degenerescence of nodule cells, and be interpreted as an example of Partner Fidelity-Feeback mechanism occurring at the level of individual cells ^41^” (Discussion section).

Figure 1, Figure 3: I understand fix^+^ and fix^-^ bacteria, but I am confused about fix+ and fix^-^ nodule. What is the definition?

A Fix^+^ (or Fix^-^) nodule is a nodule containing Fix^+^ (or Fix^-^) bacteria. We have now made this clear in the Y axis of Figure 1 and Figure 3 and in the legend of Figure 1.

Figure 2: I am not convinced: there are no data points before dpi 14, and so growth rate cannot be credibly estimated.

Indeed, we did not collect nodules before 14dpi, because most nodules are very small before 14dpi and there is a risk that sterilization agents damage them and kill nodule bacteria. We agree that this prevents us from obtaining a precise estimation of growth rate and from detecting any potential cost for nitrogen fixation (see discussion above). Nevertheless, we think that a precise estimation of growth rate is not critical for the purpose of our model, since we don’t use parameter values to extrapolate on potential mechanisms responsible for the observed dynamics. We are essentially interested in the population sizes of the different bacteria, and in their capacity to re-infect host plants (on a fine time-step, allowing to take into consideration stochastic events) over multiple inoculation cycles.

Reviewer #2:This is a significant contribution to the field of rhizobia-legume symbiosis, demonstrating nicely that non-fixing rhizobia are penalized both at the whole nodule level as well as within nodules – a matter of much debate! – and going farther to considering the time-course of this process. The modeling is a nice addition, though not necessarily required for this to be a strong paper. The multigenerational experiments seal the deal, demonstrating that the phenomenon measured within a generation translates to the expected shifts in allele frequencies over time.I have only two concerns:First, as this is something I worry about a lot in my own work, to what extent does the surface sterilization process penetrate and kill rhizobia within Mimosa nodules? This could have dramatic effects on the rhizobial populations observed in single-strain nodules that vary in size, which typically correlates with fixation status.

There is indeed a risk that the sterilization agents penetrate Mimosa nodules especially when the nodules are very small. That is why we did not collect very small nodules and sectioned the root far (ca 2mm) from the basis of each nodule (this last information is now added in Materials and methods section).

Although Fix^+^-containing nodules and Fix^–^containing nodules grow differently (Mimosa indeed form indeterminate nodules), both have a thick cortex as observed by cytology and may not be differently impacted by the sterilization. In co-infected nodules, only Fix^-^ strains are sanctioned and the relative Fix^+^/Fix^-^ fitness is even higher than when comparing single-occupied nodules.

Second, the authors claim in the text that the model fits their experimental data very well, but do not justify this statistically. In fact, looking at Figure 2 and Figure 10, the model doesn't appear to fit all that well for particular timepoints and there doesn't seem to be all that much power in rejecting the model, if this was in fact the intent. Some consideration of what the purpose of the model is would be warranted–I think it is a nice conceptual addition that shows qualitatively that cycle length and plant population size impact the evolution of fix^+^ from low frequency, but that the quantitative fit is being overstated.

Thanks for pointing this out. We agree with this comment and certainly need to reword our arguments here. The model was developed as a means to extend the analysis and provides a qualitative estimation regarding the outcome of longer and increased number of cycles, and using plant populations of different sizes. We have incorporated a “nodulation” mechanism within the model to bring it closer to this particular system. However, at its heart the model remains a stochastic dynamical system with inherent variability that has been reduced by parameterizing it on experimental data. We have rephrased our claims as to the use of the model in the manuscript as following:

Subsection “Eco-evolutionary dynamics of N2-fixers and non-fixers through serial nodulation cycles”: We have added ‘While the model is developed as a proof-of-concept, instead of a simple deterministic model we chose to include stochasticity in the nodulation process in order to reflect the variability observed in the experimental data’.

Subsection “Eco-evolutionary dynamics of N2-fixers and non-fixers through serial nodulation cycles”: ‘Simulation outcomes qualitatively matched the dynamics’.

Subsection “Eco-evolutionary dynamics of N2-fixers and non-fixers through serial nodulation cycles”: ‘accuracy’ replaced by ‘predictions’.

Subsection “Eco-evolutionary dynamics of N2-fixers and non-fixers through serial nodulation cycles”: ‘as predicted by the model’ replaced by ‘similar to what observed with the model.’

Subsection “Eco-evolutionary dynamics of N2-fixers and non-fixers through serial nodulation cycles”: ‘Data from 21 day-cycle experiment also matched the model prediction (Figure 10)’ replaced by ‘Under 21 day-cycles, both simulations and experiments lead to a slower progression of Fix^+^ subpopulations (Figure 10)’.

Discussion section: “Modelling” was replaced by “Our qualitative model”.

Discussion section: “predicted” was replaced by “A general outcome of the model”.

Discussion section: We added: “The model helps explore further combinations of number of cycles, cycle lengths and plant pool sizes to hypothesize the expected time when Fix^+^ phenotypes arise.”

Reviewer #3:

This paper presents some important data, although some of the interpretation is questionable. The important results are:1) Host sanctions against a nonfixing rhizobial strain were much more severe than previously reported for other host species (Figure 1). In the real world, however, nonfixing nodulating strains like that used in the experiments are rare, relative to less-beneficial strains that fix some nitrogen.

We agree that in nature the capacity to fix nitrogen with a legume partner can vary from no N2 fixation to levels equivalent to nitrogen-fed controls. This can result from deficiency in nitrogen fixation per se or deficiency in a previous stage, such as bacteroid differentiation or persistence, which involve different plant responses (Berrabah et al., 2015). For example, different *S. meliloti* mutants of bacA, involved in terminal differentiation, are differently impaired in N2 fixation (LeVier and Walker, 2001). Some strains of *S. meliloti* produce peptidases (HrrPs) degrading plant peptides (NCRs) involved in bacteroid differentiation (Price et al., 2015). The bactericidal activity of other plant peptides (NFS1 NFS2) affects the intracellular survival of specific bacterial genotypes, and thus nitrogen fixation (Wang et al., 2017; Yang et al., 2017). Here we were interested in the impact of nitrogen fixation per se, and thus constructed strain specifically impaired in the nitrogenase system as a first step. Construction of strains producing nitrogenase systems with gradual levels of functionality is challenging. Moreover, we can imagine that, following natural HGT of nod-nif genes in a soil bacterium, a few mutations can shift the recipient genome from Fix^-^ to full Fix^+^ by integrating nif genes in N2 and O2 regulatory circuitries (and here we show that such mutations will increase in frequency in appropriate conditions). In an experimental evolution approach using a plant pathogen having acquired a rhizobial plasmid as ancestor, one mutation was enough to convert the Nod^-^ ancestor into a Nod^+^ strain (Marchetti et al., 2010), and two mutations were enough to convert the Nod^+^ clone into a strain able to nicely intracellularly infect nodules (Capela et al., 2017).

Furthermore, there was no evidence that the resources the nifH strain spared by not fixing nitrogen are automatically invested in rhizobial fitness.

Indeed.

Therefore, we cannot conclude (subsection “Evidence for a spatial and temporal control of mutualism in Mimosa nodules”) that "low quality rhizobial partners rarely benefit from low investment in mutualism." (If there were a consensus on this, as claimed, this paper would merit publication only if it challenged that consensus.) To test that hypothesis would require comparing two nitrogen-fixing strains, with one of them providing less benefit via a mechanism that would plausibly enhance its own fitness. Examples include strains that (once inside nodules) block nodulation by other rhizobia (Tatsukami, 2016) or strains that divert more resources from N2 fixation to their own reproduction. Nonfixing mutants would not qualify.

A recent meta-analysis (Friesen, 2012) showed that there is no unequivocal example where rhizobial strains increase their own fitness (again, here we are talking about in planta fitness, i.e. ‘cheating outcome’ – see above answer to reviewer 1) while providing less nitrogen to the host, and thus concluded that “low quality partners rarely benefit from low investment in mutualism”. We acknowledge that this idea is debated. We thus modified the sentence into “The absence of a social dilemma provides support for the emerging idea that low quality rhizobial partners rarely benefit from low investment in mutualism” and moved this sentence to the Discussion section. Moreover, we think that our paper brings an additional level of understanding in this process by providing a detailed spatio-temporal analysis of rhizobial fitness.

We do agree that, in our case, we have no evidence that nifH mutants can invest more resources in their own fitness (see discussion on the cost of nitrogen fixation above) and that the suggestion of the reviewer is an attractive experiment. It would indeed help tease out the threshold in nitrogen fixation below which bacterial fitness declines and identify potential fitness gain of lower-quality partners. Theoretically this could be implemented by assuming a better reproductive success for partial fixers than total fixers. Unfortunately, we do not have such isogenic strains to conduct any confirmatory experiments. As far as we understand, the Gib- mutant strain described by Tatsukami, 2016, although fixing less nitrogen, is seemingly not fitter than the WT strain (again from a ‘cheating outcome’ perspective; see Figure 5), and therefore would not qualify for such an experiment.

In summary, we acknowledge that interactions occurring in natural ecosystems, between genetically diverse rhizobia and host plants, can encompass a wide range of subtle variations with regards to bacterial nitrogen-fixation output and cost/benefit ratio. However, our aim here was to use a reductionist approach to dissect the response of a legume plant to strains deficient in nitrogen fixation per se and not in previous symbiotic stages. Our hope is that the knowledge gained here can be then be used to interpret results from more complex (possibly ‘realistic’) interactions, in order to contribute to a broader and deeper understanding of the legume-rhizobia mutualism.

2) Sanctions were apparently imposed on a nonfixing strain within mixed nodules (Figure 1). This is arguably the most-important result, so should be featured in the Abstract, along with the "clear sectoring" (subsection “Evidence for a spatial and temporal control of mutualism in Mimosa nodules") that is presumably key to within-nodule discrimination by the host.

We have now mentioned in the Abstract that sanctions are imposed only on non-fixers in mixed nodules. This result has also now been highlighted in the Discussion section.

3) Figure 3 shows no partner choice (host-imposed discrimination during the infection process). This is consistent with past work comparing isogenic strains, but the authors should recognize the possibility of adaptive partner choice by some host plants among some real-world strains whose signals are correlated with their relative benefits.

We mentioned in the Introduction the existence of partner choice mechanisms in the rhizobium-legume symbiosis. However, to our knowledge there is no study demonstrating that nitrogen fixation per se is controlled via a partner choice mechanism.

Given these empirical results, it's obvious that fix^+^ rhizobia could invade a fix^-^ population under real-world conditions. I don't really see the point to modeling cases with only 20 plants or where plants only live 14 days after rhizobial infection, as either of those would lead to extinction of the host. If the authors want to explore "ecological factors", as claimed, they should explore some more-realistic factors like temperature or soil nitrogen. What about competition between a strain that fixes half as much nitrogen as another, diverting the resources saved to its own reproduction? That's more realistic than a nonfixing strain. Their relative fitness would depend on how the host's threshold for imposing sanctions and on the extent to which resources diverted from nitrogen fixation actually enhance rhizobial fitness. Until we have those data, modeling is almost pointless. If space in the journal is not an issue, they could keep the modeling work, but the Abstract should be rewritten to be less vague and to highlight the empirical results. One sentence in the Abstract on modeling would be plenty.

The model is useful to build hypotheses regarding the effect of parameter values that are currently beyond the reach of the experiment. Furthermore, it is a stochastic model, bringing the inherent variability of the nodulation process into consideration. Although simple, we see that the model performs well qualitatively, suggesting that it encapsulates the major factors affecting bacterial population dynamics during serial inoculation cycles.

Modeling the expansion/extinction of fixers/non-fixers in “low” conditions such as a plant population size of 20 plants is interesting for experimental evolution approaches, as mentioned in the Discussion section. In our lab we are currently evolving 18 parallel lines of bacterial populations on 20 plants each (which means 320 plants in each cycles). It is thus interesting to know when we have to collect the nodules (length of the nodulation cycle) to get better chance to select Fix^+^ mutants if they happen to arise.

In this paper we are interested in the impact of nitrogen fixation per se. We agree that it would be interesting to compare strains with different N2-fixing abilities, which would result from different capacities of the nitrogenase system and not from other traits (such as infection or persistence that of course affect the ultimate nitrogen fixation trait; see also discussion above). This imposes to construct such strains, which is challenging.

We have modified the abstract according to reviewer’s comments.

[Editors' note: further revisions were requested prior to acceptance, as described below.]

Thank you for your revision, but we are returning the revision without further review, because we don't think that you have adequately addressed the concerns about the "costs" that play a central role in your modeling effort (or we were not sufficiently clear in our previous decision letter). In your model, the "cost of nitrogen fixation" being zero strains credulity unless you are very specific about what that these costs mean.

In the revised manuscript we used the term "cost" on two occasions:

– In the Results section where we spoke of the "metabolic cost of nitrogen fixation" referring to the energetic cost paid by bacteria to fix nitrogen, independently from any kind of sanction exerted by the plant. We have now specified this in the text by writing “the metabolic cost paid by bacteria to fix nitrogen.”.

– In the Materials and methods section, where we defined the parameter "c" as being the cost of nitrogen fixation in our model. We wrote " In our simulations, the cost of nitrogen fixation c was set to 0 since we did not detect any difference in the populations of Fix^-^ or Fix^+^ nodule bacteria at 14 dpi" and “We emphasize that our definition of ‘cost of nitrogen fixation’ therefore does not refer to the actual metabolic cost of this process".

We acknowledge that this second and different definition of the "cost of nitrogen fixation" (c) in the Materials and methods is not the same as the one in the main text and agree that it is not satisfactory. We have now requalified "c" as being the actual "cost of metabolic nitrogen fixation” (same definition than in the results) and splitted the plant sanctions in two terms: *s_1_* that describes basal plant sanctions that compensate for the cost of nitrogen fixation (the only form of sanction during the initial phase of the interaction), and s_2_u_ds_(t) as an additional form of sanctions that occurs in the later phase of the interaction (Materials and methods section). We specified that “In our simulations, we set c = s_1_ since we experimentally did not detect any difference in the populations of Fix^-^ or Fix^+^ nodule bacteria at 14 dpi”. Beyond this time point, additional plant sanctions (possibly including partner fidelity-feedback) against the Fix^-^ bacteria were given by s_2_u_ds_(t)” (Materials and methods section).

Hence, we think that you need to be very specific about how these costs are described; otherwise the modelling effort could be criticized as being a tautology. If we understand your arguments correctly, the "post-PFF/post-sanctioning costs" need to be very small or even negative (benefit of cooperation) for cooperation to persist. Hence the text should be revised to say something like "the cost of cooperation is difficult to determine because such determination requires a well-mixed environment to be free of PFF. Regardless, this cost must have been overcome by PFF and/or sanctioning mechanisms. The cost we refer to is post-PFF/post-sanctioning cost of cooperation, which should be zero or negative, otherwise the mutualisms would have failed".

In the Results section we have added the following sentences: “…non-fixers did not proliferate better than fixers even at 14 dpi (Figure 1) possibly because the metabolic cost paid by bacteria to fix nitrogen is too low to be detected in our experimental conditions, or because plant sanctions/PFF and the cost of nitrogen fixation equilibrate until sanctions become prominent. The net cost of cooperation, which is the weighted cost of nitrogen fixation by any form of plant control, thus appeared to be zero or negative, preventing mutualism to fail.”.

In the Materials and methods section we have added: “The metabolic cost of nitrogen fixation is difficult to determine because this process only occurs in plant cells and thus cannot be uncoupled from potential plant control mechanisms”.

In addition, we have added in the Discussion section: “The most likely explanation is that the plant exerts a post-infection control of N_2_-fixation that overcomes the metabolic cost of nitrogen fixation paid by mutualistic bacteria”. The idea “The cost we refer to is post-PFF/post-sanctioning cost of cooperation, which should be zero or negative, otherwise the mutualisms would have failed” is once more given by the sentence: “Since control mechanisms prevent social dilemma – i.e. the possibility that one partner increases its own fitness by decreasing its investment in mutualism – and help cooperation persist, non-fixers do not threaten mutualism in our system”.

[Editors' note: further revisions were requested prior to acceptance, as described below.]

Summary:Essential revisions:1) Since their most interesting result was apparent sanctions within mixed nodules, they should cite this just-published paper: http://www.amjbot.org/citmgr?gca=amjbot;ajb.1700165v1

Done (Discussion section).

2) You will need to explain/contrast sanctions and PFF clearly in Intro to give readers sufficient background.

A description of partner choice, post-infection sanctions and PFF is now given in the Introduction.

3) A clarification of "cost".

A description of partner choice, post-infection sanctions and PFF is now given in the Introduction.

Reviewer #2:Cost definitions are key. In the model, the cost is necessarily a fitness cost because it is translating fixation rate into population growth rate (i.e., per capita fitness). This is, as articulated in the response to reviewers’ letter, completely distinct from the actual metabolic cost of nitrogen fixation in terms of ATP and reducing power. I don't see the point is having separate terms for fitness cost of fixation (c) and then the fitness effect of sanctions (s1) since these are tied together in this system anyways. It would be clearer to just have a "net fitness effect" of fixation ability that could just be zero, because the growth rate of the fix^+^ and fix^-^, is the same early in the interaction. The distinction between early phase vs late-phase seems arbitrary.

We apologize for the remaining confusion regarding the definition of the term “cost”. We have now clearly distinguished and defined the “net fitness cost” and the “metabolic cost” in the text, the Materials and methods section and Supplementary file 1:

“[T]he metabolic cost paid by bacteria to fix nitrogen in terms of ATP and reducing power” (subsection “Evidence for a spatial and temporal control of mutualism in Mimosa nodules”).

“The resulting net fitness cost of cooperation, which is the weighted metabolic cost of nitrogen fixation by any form of plant control…” (subsection “Evidence for a spatial and temporal control of mutualism in Mimosa nodules”).

“We emphasize that a net fitness cost of 0 does not necessarily imply that nitrogen fixation does not impose a metabolic burden on the bacteria” (referred to as ‘metabolic cost’ in the Results section). “Instead, this burden, if significant during the early steps of the interaction, may be compensated for by plant control mechanisms acting at a basal level” (Materials and methods section).

To further address the comment of reviewer #2, we suppressed the parameter ‘s1’ from our model and we went back to our original notation, using ‘c’ as a parameter for the net fitness cost. Therefore, we no longer have a parameter specifically referring to the ‘metabolic cost’ in our model. However, we think it is useful to maintain the two definitions in the main text, in order to justify the value of net fitness cost c=0 (subsection “Mathematical model and simulations”). This is intended to address the comment from reviewer #1 during the first revision, who pointed out that setting a cost to 0 for nitrogen fixation was confusing.

Related to this, when do Mimosa nodules really start fixing nitrogen? Prior to this point one wouldn't expect there to be a difference between fix^+^ and fix^-^ since the trait isn't expressed. In Figure 7, it isn't clear whether the first time-point (7d) is significantly different from zero – this would be worth presenting.

We have now clarified this issue by including two negative controls in Figure 7.e. the measurement of ethylene present in the acetylene reactive and in plants inoculated with a Fix^-^ strain. Statistical analysis showed that the ethylene reduced by Fix^-^ bacteria is significantly different from the negative control background from 10 dpi.

Subsection “Evidence for a spatial and temporal control of mutualism in Mimosa nodules”: Should specify here that you mean the fitness cost;

Done (now Subsection “Evidence for a spatial and temporal control of mutualism in Mimosa nodules”).

Also, the double negative "preventing mutualism to fail", is confusing and should be revised to "enabling mutualism to spread".

Done (Subsection “Evidence for a spatial and temporal control of mutualism in Mimosa nodules”).

It should be clarified which cells are contributing to the population when nodules are crushed and plated – are these a mix of former bacteroids (the ~20% that are supposedly culturable–so what is happening to the other 80%) and cells from the infection threads? If most of the cells come from infection threads, this could dilute the effect of plant control unless the plant is able to also regulate cells in ITs on the basis of the neighboring bacteroids… Please clarify what you think is happening in this system, as this could guide the next steps in dissecting this phenomenon.

Nodule bacterial population include bacteria in infection threads and bacteroids within nodule cells of the infection and fixation zones. It has been shown that only 20% of the bacteroids of the nitrogen fixation zone can resume growth on plate (Marchetti et al., 2011). The other 80% are not able to resume growth on plates for unknown reasons.

On nodule sections, Fix^-^ bacteroids of the fixation zone appeared dead at 35dpi (see Figure 5), indicative of a strong control in this plant compartment. Intracellular bacteria of the infection zone often appeared still alive at 35 dpi (Figure 5). Bacteria recovered at 35dpi from nodules only colonized by Fix^-^ bacteria (ca 5 10^[21]^) may thus represent bacteria present in the infection threads and infection zone. This may explain why the uncooperative population does not become extinct within nodules.

This is now clarified in the text (subsection “Evidence for a spatial and temporal control of mutualism in Mimosa nodules” and the Discusssion section) and in Figure 5.

Subsection “Eco-evolutionary dynamics of N2-fixers and non-fixers through serial nodulation cycles”: Should specify that you are qualitatively testing the model.

Done (subsection “Evidence for a spatial and temporal control of mutualism in Mimosanodules”):

Discussion section: degeneration

Done.

Discussion section: dashes don't match

Done.

Methods: tetracyclin –> tetracyclineDone.Reviewer #3:I continue to favor publishing the empirical results, while remaining concerned about the distinction between metabolic cost and opportunity cost.If, as the authors state that in their response, they "have no evidence that nifH mutants can invest more resources in their own fitness," then their results are only relevant to cases where there is no opportunity cost (in terms of potential rhizobial fitness, not just metabolism) to fixing nitrogen. For the Abstract to accurately describe their results, it would have to read something like "rare fixers will invade a population of nonfixing bacteria that lack mechanisms to divert resources from nitrogen fixation to their own fitness…" There's plenty of fluff in the Abstract that could be cut to include this key qualification.

We do not have any experimental evidence that nifH mutants divert resources from nitrogen fixation to their own fitness, even if it is likely. Neither have we evidence that nifH mutants do not divert these resources to their own fitness.

We thus prefer to stay factual and only use the term ‘non-fixer’ that accurately describes our strain.

Similarly, where they speculate that "the metabolic cost paid by bacteria to fix nitrogen is too low to be detected" the issue isn't metabolic cost but opportunity cost. Even if metabolic cost is very high, the resources are supplied by the plant. And, if diverting some of those resources to rhizobial reproduction isn't possible, then metabolic costs are irrelevant to rhizobial fitness. That is apparently the case for their nifH mutant. The lack of difference at 14 dpi would only be "surprising" if nifH mutants were able to divert resources to their own reproduction, prior to the imposition of sanctions.

We removed ‘Surprisingly’ (subsection “Evidence for a spatial and temporal control of mutualism in Mimosanodules”) from the text.

The real question, though, is whether strains that fix less nitrogen and can divert resources saved to their own reproduction can out-compete strains that fix more nitrogen. If we accept the importance of this question, then the empirical results merit publication, because the sanctions shown in Figure 1 are severe enough to outweigh any likely fitness benefits of diverting resources from nitrogen fixation. They don't have any data on the fitness effects of fixing less nitrogen, rather than none – this might not trigger sanctions – but that's not a reason not to publish. It's just a reason not to over-generalize in Abstract and main text. For example, in the Discussion section "in our system" is too vague, especially when coupled with generalizations about "the absence of a social dilemma." That would be a good place to point out that results could be different for "cheaters" (as opposed to "losers"), that is, strains that fix some nitrogen, but divert more resources to their own reproduction than other strains do.

We acknowledge that in nature all intermediates between non-fixing and fully N-fixing strains occur and that their fate may be different. We have added the following sentence (Discussion section):

“Yet the fate of strains able to fix intermediate levels of nitrogen fixation may be different. Monitoring the fitness of strains varying in their nitrogen fixation capacity would provide a more complete picture of mutualism control”.

Moreover, in the following line, we replaced ‘The absence of social dilemma provides support for…” by “Nevertheless, our results provide an additional example…” (Discussion section).

I accept the argument that a model that only applies to small experimental evolution studies (because of low plant numbers) is potentially useful, but the reference to "ecological factors" in the Abstract promises too much.

Although our experiments are performed in the lab, the factors we tested, i.e. the number of plants, the inoculum size and the duration of the interaction, are in essence ecological, as far as we understand the definition of “ecology”: “Ecology is the scientific analysis and study of interactions among organisms and their environment […] Ecology includes the study of interactions that organisms have with each other, other organisms, and with abiotic components of their environment” (Wikipedia).

This term is widely used for interactions studied in the laboratory, e.g. the experimental evolution of *E. coli* in minimal medium, virus-host or prey-predator systems:

Le Gac et al. Ecological and evolutionary dynamics of coexisting lineages during a long-term experiment with Escherichia coli. PNAS 2012. 109:9487-9492.

Lenski. Experimental evolution and the dynamics of adaptation and genome evolution in microbial populations. ISME J. 2017. 11:2181-2194.

Dennehy JJ. Am. Nat. 2006 Viral Ecology and the Maintenance of Novel Host Use.

Yoshida et al Nature 2003 Rapid evolution drives ecological dynamics in a predator-prey system